# IDEMPOTENT GENERATIVE NETWORK

Assaf Shocher[1,2]   Amil Dravid[1]   Yossi Gandelsman[1]   Inbar Mosseri[2]   Michael Rubinstein[2]   Alexei A. Efros[1]

[1]UC Berkeley         [2]Google Research

## ABSTRACT

We propose a new approach for generative modeling based on training a neural network to be idempotent. An idempotent operator is one that can be applied sequentially without changing the result beyond the initial application, namely $f(f(z)) = f(z)$. The proposed model $f$ is trained to map a source distribution (e.g, Gaussian noise) to a target distribution (e.g. realistic images) using the following objectives: (1) Instances from the target distribution should map to themselves, namely $f(x) = x$. We define the target manifold as the set of all instances that $f$ maps to themselves. (2) Instances that form the source distribution should map onto the defined target manifold. This is achieved by optimizing the idempotence term, $f(f(z)) = f(z)$ which encourages the range of $f(z)$ to be on the target manifold. Under ideal assumptions such a process provably converges to the target distribution. This strategy results in a model capable of generating an output in one step, maintaining a consistent latent space, while also allowing sequential applications for refinement. Additionally, we find that by processing inputs from both target and source distributions, the model adeptly projects corrupted or modified data back to the target manifold. This work is a first step towards a "global projector" that enables projecting any input into a target data distribution.

## 1 INTRODUCTION

> GEORGE: *You're gonna "overdry" it.*
> JERRY: *You, you can't "overdry."*
> GEORGE: *Why not?*
> JERRY: *Same as you can't "overwet." You see, once something is wet, it's wet. Same thing with dead: like once you die you're dead, right? Let's say you drop dead and I shoot you: you're not gonna die again, you're already dead. You can't "overdie," you can't "overdry."*

> — *"Seinfeld", Season 1, Episode 1, NBC 1989*

Generative models aim to create synthetic samples by drawing from a distribution underlying the given data. There are various approaches such as GANs (Goodfellow et al., 2014), VAE (Kingma & Welling, 2022), diffusion models (Sohl-Dickstein et al., 2015; Ho et al., 2020), pixel autoregressive methods (van den Oord et al., 2017; 2016b;a) and some recent like consistency models (Song et al., 2023) and Bayesian flow networks (Graves et al., 2023). Inputs to these models could range from samples of random noise to specific input images in conditional setups, which are then mapped to outputs aligned with a given target distribution, typically the manifold of natural images. However, each model is specifically trained to expect a particular type of input. What if we wanted a single model to be able to take any type of input, be it corrupted instances (e.g., degraded images), an alternative distribution (e.g., sketches), or just noise, and project them onto the real image manifold in one step, a kind of "Make It Real" button? As a first step toward this ambitious goal, this work investigates a new generative model based on a generalization of projection — Idempotence.

An idempotent operator is one that can be applied sequentially multiple times without changing the result beyond the initial application, namely $f(f(z)) = f(z)$. Some real-life actions can also be considered idempotent, as humorously pointed out by Jerry Seinfeld (1). One mathematical example is the function mapping $z$ to $|z|$; applying it repeatedly yields $||z|| = |z|$, leaving the result unchanged. In the realm of linear operators, idempotence equates to orthogonal projection. Over $\mathbb{R}^n$, these are matrices $A$ that satisfy $A^2 = A$, with eigenvalues that are either 0 or 1; they can be interpreted as geometrically preserving certain components while nullifying others. Lastly, the

identity function naturally exhibits idempotent behavior, as applying it multiple times leaves the input unchanged.

We propose Idempotent Generative Networks (IGN), a model based on the idea of projection. Given a dataset of examples $\{x_i\}_{i=1}^N$, Our goal is to "project" our input onto the target distribution $\mathcal{P}_x$ from which $x_i$'s are drawn. Fig. 1 illustrates the basic objectives. We assume that distributions $\mathcal{P}_z$ and $\mathcal{P}_x$ lie in the same space. Given that, it is valid to apply $f$ to a given example $x \sim \mathcal{P}_x$. What should the outcome of doing so be then? The natural answer to that is "nothing". Considering the intuition of projection, an instance that already lies on the target manifold should just remain the same- "You can't overdry". The

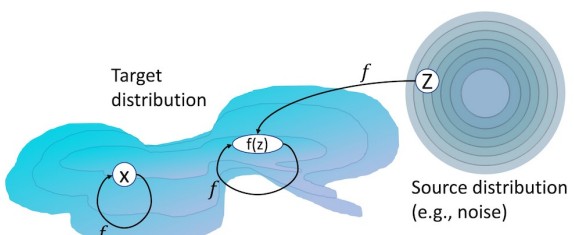

Figure 1: The basic idea behind IGN: real examples $(x)$ are invariant to the model $f$: $f(x) = x$. other inputs $(z)$ are projected onto the manifold of instances that $f$ maps to themselves by optimizing for $f(f(z)) = f(z)$.

first objective is then perfectly fulfilled when $f(x) = x$. We can leverage this notion, and define the estimated manifold of the data as the sub-set of all instances that $f$ maps close to themselves.

Next, we want to map instances from a different distribution onto that estimated manifold. To do so, we want $f(z)$ to be on the manifold for every $z \sim \mathcal{P}_z$, which is characterized by being mapped to itself. This defines our second objective, Idempotence : $f(f(z)) = f(z)$. While the aforementioned objectives ensure both $x$ and $f(z)$ reside on the estimated target manifold, they do not inherently constrain what else populates that manifold. To address this, we introduce a third term, to tighten the manifold, pushing for $f(f(z)) \neq f(z)$. The intricacy of reconciling opposing loss terms is unraveled in Section 2.1.

While the landscape of generative modeling is rich with diverse methodologies, our Idempotent Generative Network (IGN) features specific advantages that address existing limitations. In contrast to autoregressive methods, which require multiple inference steps, IGN produces robust outputs in a single step, akin to one-step inference models like GANs. Yet, it also allows for optional sequential refinements, reminiscent of the capabilities in diffusion models. Unlike diffusion models, however, IGN maintains a consistent latent space, facilitating easy manipulations and interpolations. The model shows promise in generalizing beyond trained instances, effectively projecting degraded or modified data back onto the estimated manifold of the target distribution. Moreover, the model's ability to accept both latent variables and real-world instances as input simplifies the editing process, eliminating the need for the inversion steps commonly required in other generative approaches. We draw connections to other generative models in Section 5.

## 2 METHOD

We start by presenting our generative model, IGN. It is trained to generate samples from a target distribution $\mathcal{P}_x$ given input samples from a source distribution $\mathcal{P}_z$. Formally, given a dataset of examples $\{x_i\}_{i \in \{1,\dots,n\}}$, with each example drawn from $\mathcal{P}_x$, we train a model $f$ to map $\mathcal{P}_z$ to $\mathcal{P}_x$. We assume both distributions $\mathcal{P}_z$ and $\mathcal{P}_x$ lie in the same space, i.e., their instances have the same dimensions. This allows applying $f$ to both types of instances $z \sim \mathcal{P}_z$ and $x \sim \mathcal{P}_x$.

Next, we describe the optimization objectives, the training procedure of the model, the architecture, and practical considerations in the training.

### 2.1 OPTIMIZATION OBJECTIVES

The model optimization objectives rely on three main principles. First, each data sample from the target distribution should be mapped by the model to itself. Second, the model should be idempotent - applying it consecutively twice should provide the same results as applying it once. Third, The subset of instances that are mapped to themselves should be as small as possible. Next we explain the objectives and show how these principles are translated to optimization objectives.

**Reconstruction objective.** Our first objective, as motivated in the introduction, is the reconstruction objective, which is perfectly achieved when each sample $x \sim \mathcal{P}_x$ is mapped to itself:

$$f(x) = x \tag{1}$$

Given a distance metric $D$ (e.g., $L_2$), we define the drift measure of some instance $y$ as:

$$\delta_\theta(y) = D\big(y, f_\theta(y)\big) \tag{2}$$

Where $\theta$ are the parameters of a model $f_\theta$. We then seek to minimize the drift measure:

$$\min_\theta \delta_\theta(x) = \min_\theta D\big(x, f_\theta(x)\big) \tag{3}$$

The fact that real instances are mapped to themselves motivates us to define the ideal estimated target manifold as the subset of all possible inputs that are mapped to themselves by our model:

$$\mathcal{S} = \{y : f(y) = y\} = \{y : \delta(y) = 0\} \tag{4}$$

**Idempotent objective.** We desire $f$ to map any instance sampled from the source distribution onto the estimated manifold:

$$f(z) \in \mathcal{S} \quad z \sim \mathcal{P}_z \tag{5}$$

Taking the definition in Eq. 4 and combining with our goal in 5, we can define our second objective, that when perfectly achieved we get:

$$f(f(z)) = f(z) \tag{6}$$

This implies that $f$ is *idempotent* over the domain of all possible $z \sim \mathcal{P}_z$. This idempotence objective is formulated then as follows.

$$\min_\theta \delta_\theta(f_\theta(z)) = \min_\theta D\left(f_\theta(z), f_\theta(f_\theta(z))\right) \tag{7}$$

However, we will next see that directly optimizing this formulation of Idempotnce has a caveat that requires us to split it into two separate terms.

**Tightness objective.** In the formulation of the objectives so far, there is a missing link. The reconstruction objective, if optimized perfectly (eq. 2.1), determines that all given examples are on the estimated manifold. However, it does not imply that other instances are not on that manifold. An extreme example is that if $f$ is identity $f(z) = z \quad \forall z$, it perfectly satisfies both objectives. Furthermore, the idempotent objective as formulated so far, is encouraging the manifold to expand.

Fig. 2 and fig. 17 illustrate this problem. There are two distinct pathways of gradients that flow when the idempotent objective in eq. 7 is minimized, both of which would contribute when optimizing. The first pathway, which is the desired one, is by modifying $f$ such that $f(z)$, the first application of $f$ to $z$ (red in fig. 2) is better mapped to the currently defined manifold. The second pathway is for a given $y = f(z)$ making $f(y)$ closer to $y$ (green in fig. 2). Since the estimated manifold is defined by $\mathcal{S} = \{y : f(y) = y\}$ This second way of optimizing is effectively expanding it.

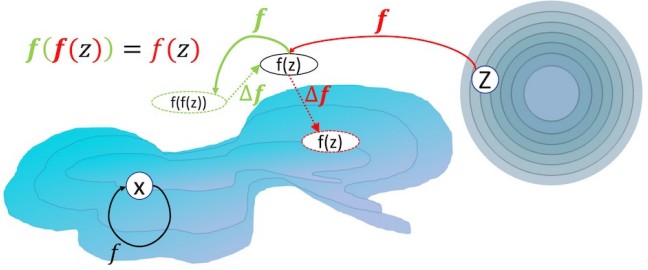

Figure 2: Two distinct pathways to enforce Idempotence: By updating $f$ so it maps $f(z)$ into $\mathcal{S} = \{y : f(y) = y\}$ (updating through first instatiation, $\Delta f$ ) or by expanding the $\mathcal{S} = \{y : f(y) = y\}$ area so that for a *given* $y = f(z)$, we get $f(y) = y$ (updating through second instantiation $\Delta f$). If we encourage the red update while discouraging the green one, we simultaneously map into the estimated manifold while tightening it around the data examples. See also fig. 17 for illustration of the gradient pathways.

In order to discourage the incentive to expand the manifold, we only optimize w.r.t. the first (inner) instantiation of $f$, while treating the second (outer) instantiation as a frozen copy of the current state of $f$. We denote by $\theta'$ the parameters of the frozen copy of $f$. They are equal in value to $\theta$ but they are different entities, in the sense that a gradient taken w.r.t $\theta$ will not affect $\theta'$.

$$L_{idem}(z;\theta,\theta') = \delta_{\theta'}(f_\theta(z)) = D\left(f_{\theta'}(f_\theta(z)), f_\theta(z)\right) \tag{8}$$

We denote the expectation of losses as

$$\mathcal{L}_{idem}(\theta;\theta') = \mathbb{E}_z\left[L_{idem}(z;\theta,\theta')\right] \tag{9}$$

Eq. 8 prevents the encouragement to expand the manifold, but we are interested in tightening the manifold as much as possible. We therefore **maximize** the distance between $f(y)$ and $y$ for a **given** $y = f(z)$. Effectively, this trains $f$ to exclude that generated $y = f(z)$ from the estimated manifold $\mathcal{S} = \{y : f(y) = y\}$ by optimizing only the second (outer) instantiation of $f$, treating the first as a frozen copy. The term we want to minimze is then

$$L_{tight}(z;\theta,\theta') = -\delta_\theta(f_{\theta'}(z)) = -D\left(f_\theta(f_{\theta'}(z)), f_{\theta'}(z)\right) \tag{10}$$

Notice that $L_{tight}(z;\theta,\theta') = -L_{idem}(z;\theta',\theta)$. This induces an adversarial fashion for training them together. However, there is no alternating. Gradients are accumulated on $\theta$ in a single step.

**Final optimization objective.** Combining the three optimization terms described above brings us to the final loss:

$$\begin{aligned}
\mathcal{L}(\theta,\theta') &= \mathcal{L}_{rec}(\theta) + \lambda_i\mathcal{L}_{idem}(\theta;\theta') + \lambda_t\mathcal{L}_{tight}(\theta;\theta') \\
&= \mathbb{E}_{x,z}\left[\delta_\theta(x) + \lambda_i\delta_{\theta'}(f_\theta(z)) - \lambda_t\delta_\theta(f_{\theta'}(z))\right]
\end{aligned} \tag{11}$$

with $\mathcal{L}_{rec}(\theta) = \mathbb{E}_x\left[D(f_\theta(x), x)\right]$ being the reconstruction term and $\lambda_i$ and $\lambda_t$ being the weights of the idempotent and tightening loss terms respectively. Note that while the losses are assigned with $\theta' = \theta$, the gradient which is made of partial derivatives is only w.r.t. the original argument $\theta$ of the loss $\mathcal{L}_{idem}(z;\theta,\theta')$. The general update rule is therefore:

$$\begin{aligned}
\theta' &\leftarrow \theta \\
\theta &\leftarrow \theta - \eta\nabla_\theta\mathcal{L}(\theta,\theta')
\end{aligned} \tag{12}$$

## 2.2 TRAINING

For a single model $f$ that appears multiple times in the calculation, we want to optimize by taking gradients of distinct losses w.r.t. different instantiations of $f$. fig. 2, eq. 13 and fig. 3 all share the same color coding. Red indicates the update of $f$ through its first (inner) instantiation, by minimizing $\delta$. Green indicates the update of $f$ through its second (outer) instantiation, by maximizing $\delta$; We examine the gradient of $\delta_\theta(f(z))$.

$$\nabla_\theta\delta(f(z)) = \underbrace{\frac{\partial\delta(f(z))}{\partial f(f(z))}\frac{df(\cdot)}{d\theta}\Big|_{f(z)}}_{\mathcal{L}_{tight}:\ \text{Gradient ascent}\uparrow} + \underbrace{\left(\frac{\partial\delta(f(z))}{\partial f(f(z))}\frac{\partial f(f(z))}{\partial f(z)} + \frac{\partial\delta(f(z))}{\partial f(z)}\right)\frac{df(\cdot)}{d\theta}\Big|_z}_{\mathcal{L}_{idem}:\ \text{Gradient descent}\downarrow} \tag{13}$$

The two terms of the gradient exemplify the different optimization goals for the different appearances of $f$. Optimizing $\mathcal{L}_{tight}$ is done by gradient ascent on the first term while optimizing $\mathcal{L}_{idem}$ is done by gradient descent on the second term.

For $\mathcal{L}_{tight}$ it is trivial to prevent optimization of the first (inner) instantiation of $f$. As depicted in fig. 3, it can be done by stopping the gradients in the backward process between the two instantiations of $f$, treating $f(z)$ as a static input. This method, however, cannot be applied for $\mathcal{L}_{idem}$. Eq. 13 shows that the gradient w.r.t the wanted first instantiation of $f$, $\frac{\partial\delta(f(z))}{\partial f(z)}$ is calculated with chain rule through the second $\frac{\partial\delta(f(z))}{\partial f(f(z))}$. To cope, fig. 3 shows that we employ a copy of the model, $f_{copy}$. It is updated at every iteration to be identical to $f$, but as a different entity, we can calculate backpropagation through it without accumulating gradients w.r.t. its parameters.

fig. 3 and source-code. 2.2 show how the training is performed in practice. For efficiency we first calculate $f(z)$ that can be shared by both the idempotent loss and the tightening loss. In source-code. 2.2 we provide the basic training PyTorch code for IGN. This is the actual code used for MNIST experiments, once provided with a model, an optimizer and a data-loader.

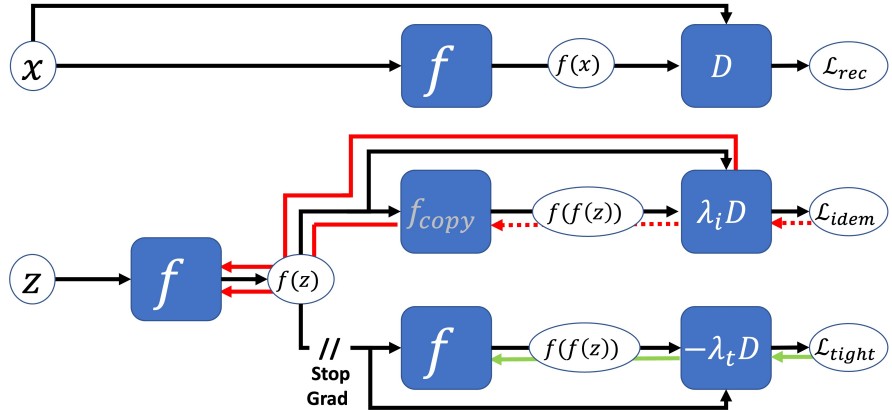

Figure 3: A diagram of the proposed method. The top depicts the reconstruction term over real data. The bottom depicts the Idempotence and tightness terms. The colored arrows depict the gradients. The colors match the colors in eq. 13 and fig. 2. Dashed arrow indicates back propagation without accumulating gradients on the parameters it passes through. The final loss is the sum of all the losses.

```python
def train(f, f_copy, opt, data_loader, n_epochs):
    for epoch in range(n_epochs):
        for x in data_loader:
            z = torch.randn_like(x)

            # apply f to get all needed
            f_copy.load_state_dict(f.state_dict())
            fx = f(x)
            fz = f(z)
            f_z = fz.detach()
            ff_z = f(f_z)
            f_fz = f_copy(fz)

            # calculate losses
            loss_rec = (fx - x).pow(2).mean()
            loss_idem = (f_fz - fz).pow(2).mean()
            loss_tight = -(ff_z - f_z).pow(2).mean()

            # optimize for losses
            loss = loss_rec + loss_idem + loss_tight * 0.1
            opt.zero_grad()
            loss.backward()
            opt.step()
```

Source Code 1: IGN training routine (PyTorch)

## 2.3 ARCHITECTURE AND OPTIMIZATION

**Network architecture.** The typical model to be used with IGN is built as an autoencoder. One possibility is using an existing GAN architecture, "flipping" the order such that the encoder is the discriminator, with the binary classification head chopped off, and the encoder is the generator.

**Tightening loss metric.** One undesirable effect caused by $\mathcal{L}_{tight}$ is that it benefits from applying big modifications even to a relatively good generated instance. Moreover, optimized to increase distance between input and output encourages high gradients and instability. To ameliorate these issues we modify the distance metric for $\mathcal{L}_{tight}$ and limit its value. We use a smooth clamp by

hyperbolic tangent with the value dependent on the current reconstruction loss for each iteration

$$L_{tight}(z) = \tanh\left(\frac{\tilde{L}_{tight}(z)}{aL_{rec}(z)}\right) aL_{rec}(z) \tag{14}$$

With $\tilde{L}_{tight}$ the loss as defined before and $a \geq 1$ a constant ratio. The rationale is that if at any given time a latent that is mapped far out of the estimated manifold, we have no reason to push it further.

**Noise distribution.** We found slight improvement occurs when instead of standard Gaussian noise we sample noise with frequency-statistics as the real data. We apply a Fast Fourier Transform (FFT) to a batch of data and take the mean and variance for the real and imaginary parts of each frequency. We then use these statistics to sample and apply inverse FFT to get the noise. Examples of how this noise looks like are shown in fig. 4.

## 3 THEORETICAL RESULTS

Under idealized assumptions, our proposed training paradigm leads to a noteworthy theoretical outcome: After convergence, the distribution of instances generated by the model is aligned with the target distribution. Moreover, the Idempotence loss describes at each step the probability of a random input $z$ to map onto the manifold estimated by the other losses.

**Theorem 1.** *Under ideal conditions, IGN converges to the target distribution.*
*We define the generated distribution, represented by $\mathcal{P}_\theta(y)$, as the PDF of $y$ when $y = f_\theta(z)$ and $z \sim \mathcal{P}_z$. We split the loss into two terms.*

$$\mathcal{L}(\theta; \theta') = \underbrace{\mathcal{L}_{rec}(\theta) + \lambda_i \mathcal{L}_{tight}(\theta; \theta')}_{\mathcal{L}_{rt}} + \lambda_t \mathcal{L}_{idem}(\theta; \theta') \tag{15}$$

*We assume a large enough model capacity such that both terms obtain a global minimum:*

$$\theta^* = \arg\min_\theta \mathcal{L}_{rt}(\theta; \theta^*) = \arg\min_\theta \mathcal{L}_{idem}(\theta; \theta^*) \tag{16}$$

*Then, $\exists \theta^* : \mathcal{P}_{\theta^*} = \mathcal{P}_x$ and for $\lambda_t = 1$, this is the only possible $\mathcal{P}_{\theta^*}$.*

*Proof.* We first demonstrate that $\mathcal{L}_{rt}$ minimizes the drift $\delta$ over the target distribution while maximizing it at every other $f(z)$. Next, we show that $\mathcal{L}_{idem}$ maximizes the probability of $f$ to map $z$ to minimum drift areas.

We first find the global minimum of $\mathcal{L}_{rt}$ given the current parameters $\theta^*$:

$$\mathcal{L}_{rt}(\theta; \theta^*) = \mathbb{E}_x\big[D(f_\theta(x), x)\big] - \lambda_t \mathbb{E}_z\big[D(f_\theta(f_{\theta^*}(z)), f_{\theta^*}(z))\big] \tag{17}$$

$$= \int \delta_\theta(x)\mathcal{P}_x(x)dx - \lambda_t \int \delta_\theta(f_{\theta^*}(z))\mathcal{P}_{\theta^*}(z)dz \tag{18}$$

We now change variables. For the left integral, let $y := x$ and for the right integral, let $y := f_{\theta^*}(z)$.

$$\mathcal{L}_{rt}(\theta; \theta^*) = \int \delta_\theta(y)\mathcal{P}_x(y)dy - \lambda_t \int \delta_\theta(y)\mathcal{P}_{\theta^*}(y)dy \tag{19}$$

$$= \int \delta_\theta(y)\Big(\mathcal{P}_x(y) - \lambda_t \mathcal{P}_{\theta^*}(y)\Big)dy \tag{20}$$

We denote $M = \sup_{y_1, y_2} D(y_1, y_2)$, where the supremum is taken over all possible pairs $y_1, y_2$. Note that $M$ can be infinity. Since $\delta_\theta$ is non-negative, the global minimum for $\mathcal{L}_{rt}(\theta; \theta^*)$ is obtained when:

$$\delta_{\theta^*}(y) = M \cdot \mathbb{1}_{\{\mathcal{P}_x(y) < \lambda_t \mathcal{P}_{\theta^*}(y)\}} \quad \forall y \tag{21}$$

Next, we characterize the global minimum of $\mathcal{L}_{idem}$ given the current parameters $\theta^*$:

$$\mathcal{L}_{idem}(\theta, \theta^*) = \mathbb{E}_z\big[D\left(f_{\theta^*}(f_\theta(z)), f_\theta(z)\right)\big] = \mathbb{E}_z\big[\delta_{\theta^*}(f_\theta(z))\big] \tag{22}$$

Plugging in Eq. 21 and substituting $\theta^*$ with $\theta$ as we examine the minimum of the inner $f$:

$$\mathcal{L}_{idem}(\theta; \theta^*) = M \cdot \mathbb{E}_z\big[\mathbb{1}_{\{\mathcal{P}_x(y) < \lambda_t \mathcal{P}_\theta(y)\}}\big] \tag{23}$$

To obtain $\theta^*$, according to our assumption in Eq. 16, we take $\arg\min_\theta$ of Eq. 23:

$$\theta^* = M \cdot \arg\min_\theta \mathbb{E}_z\big[\mathbb{1}_{\{\mathcal{P}_x(y) < \lambda_t \mathcal{P}_\theta(y)\}}\big] \tag{24}$$

The presence of parameters to be optimized in this formulation is in the notion of the distribution $\mathcal{P}_\theta(y)$. If $\mathcal{P}_{\theta^*} = \mathcal{P}_x$ and $\lambda_t \leq 1$, the loss value will be 0, which is its minimum. If $\lambda = 1$, $\theta^* : \mathcal{P}_{\theta^*} = \mathcal{P}_x$ is the only minimizer. This is because the total sum of the probability needs to be 1. Any $y$ for which $\mathcal{P}_\theta(y) < \mathcal{P}_x(y)$ would necessarily imply that $\exists y$ such that $\mathcal{P}_\theta(y) > \mathcal{P}_x(y)$, which would increase the loss. □

Qualitatively, the value $\delta_\theta(y)$ can be thought of as energy, minimized where the probability $\mathcal{P}_x(y)$ is high and maximized where $\mathcal{P}_x(y)$ is low. Under the ideal assumptions, it is binary, but in practical scenarios, it would be continuous.

Interestingly, Eq. 23 returns 0 if $\delta_\theta(y) = 0$ which is the definition of being on the estimated manifold. This indicator essentially describes the event of $f_\theta(z) \notin \mathcal{S}_\theta$. Taking the expectation over the indicator yields the probability of the event. The loss is the probability of a random $z$ to be mapped outside of the manifold. Optimizing idempotence is essentially **maximizing the portion of $z$'s that are mapped onto the manifold**.

In practice, we use $\lambda_t < 1$. While the theoretical derivation guarantees a single desired optimum for $\lambda_t = 1$, the practical optimization of a finite capacity neural network suffers undesirable effects such as instability. The fact that $f$ is continuous makes the optimal theoretical $\theta^*$ which produces a discontinuous $\delta_{\theta^*}$ unobtainable in practice. This means that $\mathcal{L}_{tight}$ tends to push toward high values of $\delta_\theta(y)$ also for $y$ that is in the estimated manifold. Moreover, in general, it is easier to maximize distances than minimize them, just by getting big gradient values.

## 4 Experimental Results

Following the training scheme outlined in Sections 2.2 and 2.3, we train IGN on two datasets - CelebA and MNIST. We present qualitative results for the two datasets, as well out-of-distribution projection capabilities and latent space manipulations.

Our generative outcomes, at this stage, are not competitive with sate-of-the-art models. Our experiments currently operate with smaller models and lower-resolution datasets. In our exploration, we primarily focus on a streamlined approach, deferring additional mechanisms to maintain the purity of the primary method. It's worth noting that foundational generative modeling techniques, like GANs Goodfellow et al. (2014) and Diffusion Models Sohl-Dickstein et al. (2015), took considerable time to reach their matured, scaled-up performance. We view this as a preliminary step, providing initial evidence of the potential capabilities. Our future work will aim at refining and scaling up the approach.

**Experimental settings.** We evaluate IGN on MNIST (Deng, 2012), a dataset of grayscale hand-written digits, and CelebA (Liu et al., 2015), a dataset of face images. We use image resolutions of $28 \times 28$ and $64 \times 64$ respectively. We adopt a simple autoencoder architecture, where the encoder is a simple five-layer discriminator backbone from DCGAN, and the decoder is the generator. The training and network hyperparameters are presented in Table 1.

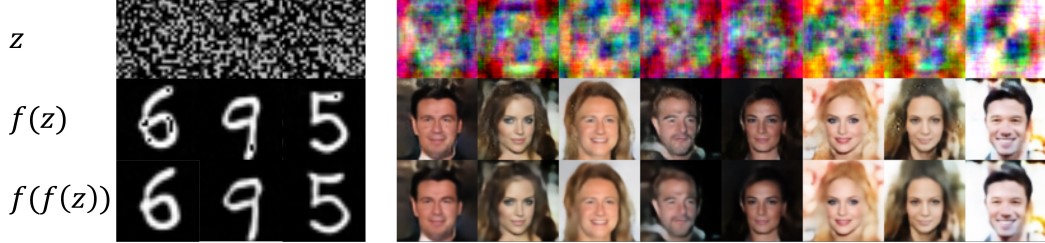

Figure 4: Examples of MNIST and CelebA IGN generations from input Gaussian noise for IGN. Notice that in some cases $f(f(z))$ corrects for artifacts in $f(z)$.

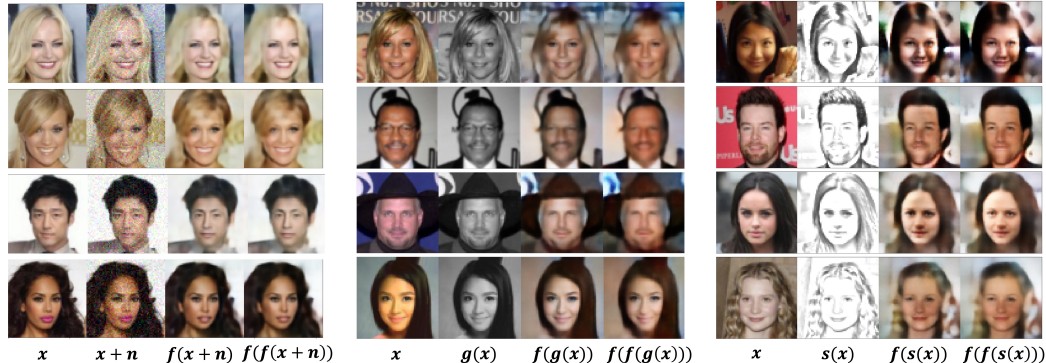

$x$     $x+n$    $f(x+n)$ $f(f(x+n))$       $x$      $g(x)$    $f(g(x))$ $f(f(g(x)))$       $x$      $s(x)$    $f(s(x))$ $f(f(s(x)))$

Figure 5: Projections of images from different distributions using IGN. We demonstrate that IGN can project noisy images $x + n$ (left), grayscale images $g(x)$ (middle), and sketches $s(x)$ (right) onto the learned natural image manifold to perform image-to-image translation. See appendix for details on the degradations.

**Generation results.** Figure 4 presents qualitative results for the two datasets after applying the model once and consecutively twice. We report FID=39 (DCGAN FID=34). As shown, applying IGN once ($f(z)$) results in coherent generation results. However, artifacts can be present, such as holes in MNIST digits, or distorted pixels at the top of the head and hair in the face images. Applying $f$ again ($f(f(z))$) corrects for these, filling in the holes, or reducing the total variation around noisy patches in the face. Figure 7 shows additional results, as well as applying $f$ three times. Comparing $f(f(f(z)))$ to $f(f(z))$ shows that when the images get closer to the learned manifold, applying $f$ again results in minimal changes. See a large uncurated collection Figures 11-14.

**Latent Space Manipulations.** We demonstrate IGN has a consistent latent space by performing manipulations, similarly as shown for GANs (Radford et al., 2015). **Latent space interpolation videos can be found in the supplementary material.** We sample several random noises, take linear interpolation between them and apply $f$. In The videos left to right: $z, f(z), f(f(z)), f(f(f(z)))$. Fig. 6 shows latent space arithmetics. Formally, we consider three inputs $z_{\text{positive}}, z_{\text{negative}}$ and $z$, such that $f(z_{\text{positive}})$ has a specific image property that $f(z_{\text{negative}})$ and $f(z)$ do not have (e.g. the faces in the two former images have glasses, while the latter does not have them). The result of $f(z_{\text{positive}} - z_{\text{negative}}) + z)$ is an edited version of $f(z)$ that has the property.

**Out-of-Distribution Projection.** We validate the potential for IGN as a "global projector" by inputting images from a variety of distributions into the model to produce their "natural image" equivalents (i.e.: project onto IGN's learned manifold). We demonstrate this by denoising noised images $x+n$, colorizing grayscale images $g(x)$, and translating sketches $s(x)$ to realistic images in Fig. 5. Although the projected images are not perfect reconstructions of the original images $x$, these inverse tasks are ill-posed. IGN is able to create natural-looking projections that adhere to the structure of the original images. As shown, sequential applications of $f$ can improve the image quality (e.g. it removes dark and smoky artifacts in the projected sketches). Note that IGN was only trained on natural images and noise, and did not see distributions such as sketches or grayscale images. While other methods explicitly train for this

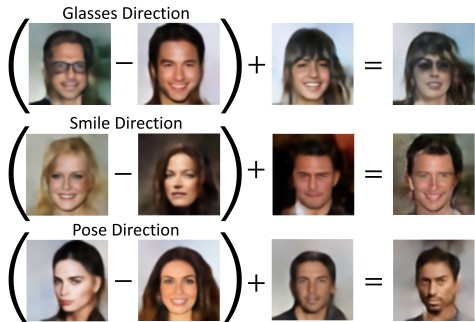

Figure 6: Input noise arithmetic. Similar to GANs, arithmetic operations can be performed in the input noise space to idempotent generative networks to find interpretable axes of variation.

task (Zhu et al., 2017; Isola et al., 2017), this behavior naturally emerges in IGN as a product of its projection objective. Moreover, due to the autoencoding architecture of IGN, we do not need to rely on inversion for editing. Instead, we rely solely on forward passes through the network.

## 5 RELATED WORK

**Generative Adversarial Networks (GANs).** IGN incorporates elements of adversarial training (Goodfellow et al., 2014), evident in the relationship between $\mathcal{L}_{\text{idem}}$ and $\mathcal{L}_{\text{tight}}$, which are negatives of each other. One could view $\delta$ as a discriminator trained using $\mathcal{L}_{\text{rec}}$ for real examples and $\mathcal{L}_{\text{tight}}$ for generated ones, while $f$ serves as the generator trained by $\mathcal{L}_{\text{idem}}$. Unique to IGN is a form of adversarial training we term *"self-adversarial"*, Put simply, $f$ is both the generator and the discriminator. This streamlined architecture affects the optimization process. Rather than alternating between two networks, a single model accumulates gradients from both discriminative and generative perspectives in each step.

**Energy Based Models (EBMs).** In Energy-Based Models (EBMs; Ackley et al. (1985)), a function $f$ is explicitly trained to serve as an energy metric, assigning higher values to less desirable examples and lower values to those that fit the model well. IGN introduces a similar, yet distinct paradigm: rather than $f$ acting as the energy function, this role is filled by $\delta(y) = D(f(y), y)$. The model trains $f$ to be idempotent, with the objective to minimize $\delta(f(z))$. A successful training procedure would align the range of $f$ with the low-energy regions as measured by $\delta$. This reduces the need for separate optimization procedures to find the energy minimum. From another perspective, $f$ can be viewed as a transition operator that maps high-energy inputs toward a low-energy domain.

**Energy Based Generative Adversarial Network (EBGAN) Zhao et al. (2017).** Combining GANs and EBMs, EBGAN is the closest existing model to IGN. EBGAN is a GAN where the discriminator is built as an autoencoder. The discriminator Like IGN, uses a reconstruction loss instead of a binary loss. The EBGAN discriminator is not trained to project as it only sees real or generated images, but not latents or out-of-distribution instances. It is trained to discriminate. IGN differs in its mentioned self-adversarialness having $f$ as both the generator and critic. The same model judges the quality while trying to improve it.

**Diffusion Models.** In both diffusion models (Sohl-Dickstein et al., 2015) and IGN, the model can be sequentially applied. Additionally, both methods train the model to transition an input along a path between a source distribution and a target data manifold. In diffusion models, this path is dictated by a predefined noise schedule. At inference, the model takes small, incremental steps, effectively performing a form of gradient descent to transition from complete noise—representing the source distribution—to the target data manifold. IGN diverges from this approach. Instead of a predefined path dictated by a noise schedule or any other set rule, the trajectory between distributions is determined solely by the model's learning process. Unlike diffusion models, IGN doesn't employ incremental gradient steps toward the data manifold. Instead, it is trained to approximate as closely as possible to the target manifold in a single step. It can be reapplied for further refinement if needed.

## 6 LIMITATIONS

**Mode collapse.** Similar to GANs, our model can experience mode collapse and is not practically guaranteed to generate the entire target distribution. Some methods attempt to overcome this failure mode in GANs (Mao et al., 2019; Durall et al., 2020). We plan to investigate if these methods are applicable to our generative model as well.

**Blurriness.** Similar to VAEs and other autoencoders, our model suffers from blurry generated samples. Although repeated applications can fix artifacts to make images appear more natural, they may also smoothen them towards an average-looking image. One possible solution to this problem is to replace the naive reconstruction loss with a perceptual loss Johnson et al. (2016). Another solution is to use a two-step approach and apply our model on latents instead of pixels (similar to Rombach et al. (2021)). We plan to investigate it in future work.

ACKNOWLEDGEMENTS

The authors would like to thank Karttikeya Mangalam, Yannis Siglidis, Konpat Preechakul, Niv Haim, Niv Granot and Ben Feinstein for the helpful discussions. Assaf Shocher gratefully acknowledges financial support for this publication by the Fulbright U.S. Postdoctoral Program, which is sponsored by the U.S. Department of State. Its contents are solely the responsibility of the author and do not necessarily represent the official views of the Fulbright Program or the Government of the United States. Amil Dravid is funded by the US Department of Energy Computational Science Graduate Fellowship. Yossi Gandelsman is funded by the Berkeley Fellowship and the Google Fellowship. Additional funding came from DARPA MCS and ONR MURI.

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

## A    VISUAL COMPARISON OF ITERATIVE APPLICATIONS OF $f$

Also see videos in supplementary material.

$f(z)$

$f(f(z))$

$f(f(f(z)))$

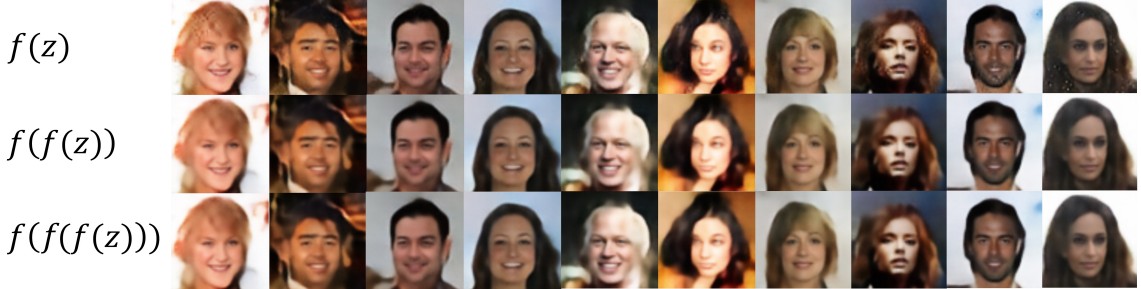

Figure 7: Comparison of iterative applications of $f$. As the generated images approach the learned manifold, sequential applications of $f$ have smaller effects on the outputs.

$f(z)$

$f(f(z))$

$f(f(f(z)))$

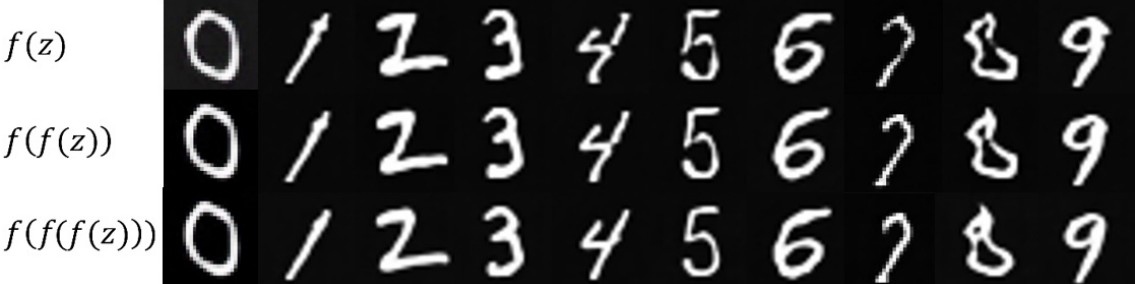

Figure 8: Comparison of iterative applications of $f$ on MNIST.

## B  MORE PROJECTIONS

$$X \qquad \tilde{X} \qquad f(\tilde{X}) \quad f(f(\tilde{X}))$$

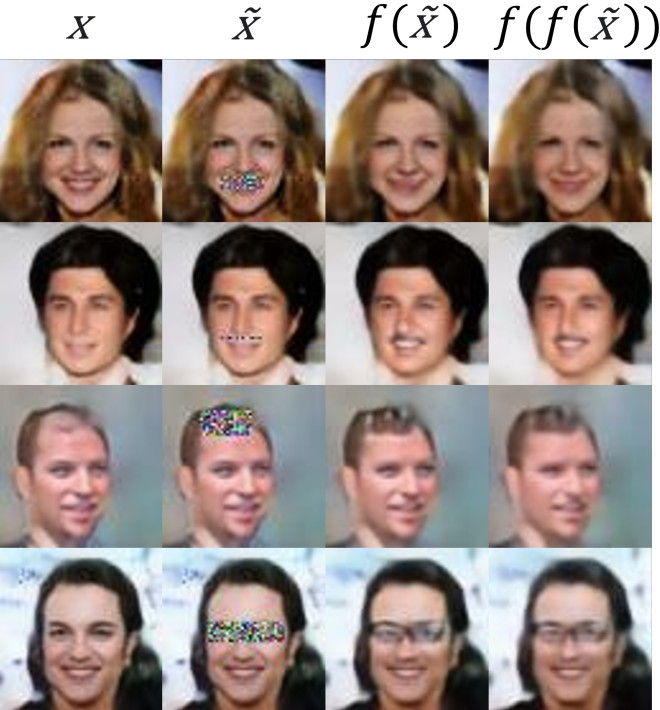

Figure 9: Projection-based edits. By simply masking out a region of interest and adding noise for stochastic variation, we can conduct fine-grained edits, such as closing the mouth, adding hair or facial hair, and putting on glasses.

$$z \qquad f(z) \qquad x \qquad x_{mask} \quad f(x_{mask})$$

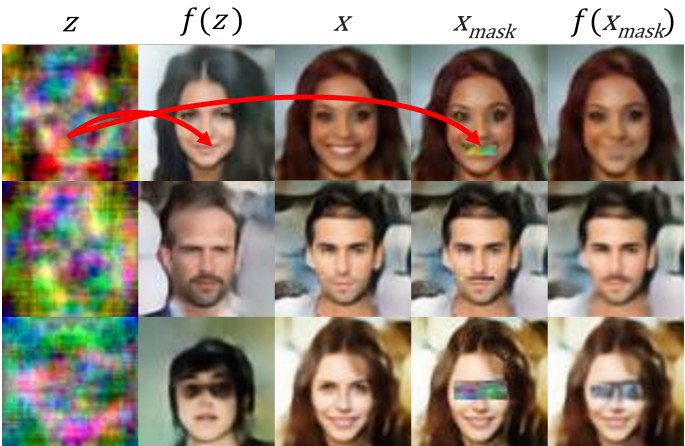

Figure 10: Projection-based compositing. Given a reference image $f(z)$, we can use the noise spatially corresponding to an attribute of interest, place it on another image $x$, and project it in order to transfer an attribute, such as glasses, facial hair, etc.

## C    IMPLEMENTATION DETAILS

| Operation | Kernel | Strides | Padding | Feature maps | BN? | Nonlinearity |
|---|---|---|---|---|---|---|
| Encoder – $3 \times 64 \times 64$ input | | | | | | |
| Convolution | $4 \times 4$ | $2 \times 2$ | 1 | 64 | $\times$ | Leaky ReLU |
| Convolution | $4 \times 4$ | $2 \times 2$ | 1 | 128 | $\checkmark$ | Leaky ReLU |
| Convolution | $4 \times 4$ | $2 \times 2$ | 1 | 256 | $\checkmark$ | Leaky ReLU |
| Convolution | $4 \times 4$ | $2 \times 2$ | 1 | 512 | $\checkmark$ | Leaky ReLU |
| Convolution | $4 \times 4$ | $1 \times 1$ | 0 | 512 | $\times$ | None |
| Decoder – $512 \times 1 \times 1$ input | | | | | | |
| Transposed Convolution | $4 \times 4$ | $1 \times 1$ | 0 | 512 | $\checkmark$ | ReLU |
| Transposed Convolution | $4 \times 4$ | $2 \times 2$ | 1 | 256 | $\checkmark$ | ReLU |
| Transposed Convolution | $4 \times 4$ | $2 \times 2$ | 1 | 128 | $\checkmark$ | ReLU |
| Transposed Convolution | $4 \times 4$ | $2 \times 2$ | 1 | 64 | $\checkmark$ | ReLU |
| Transposed Convolution | $4 \times 4$ | $2 \times 2$ | 1 | 3 | $\times$ | Tanh |
| Loss metric $D$ | $L_1$: $D(y_1, y_2) = \|y_1 - y_2\|_1$ | | | | | |
| Loss terms weights | $\lambda_r = 20, \lambda_i = 20, \lambda_t = 2.5$ | | | | | |
| $\mathcal{L}_{thight}$ clamp ratio | $a = 1.5$ | | | | | |
| Optimizer | Adam ($\alpha = 0.0001, \beta_1 = 0.5, \beta_2 = 0.999$) | | | | | |
| Batch size | 256 | | | | | |
| # GPUs | 8 | | | | | |
| Iterations | 1000 | | | | | |
| Leaky ReLU slope | 0.2 | | | | | |
| Weight, bias initialization | Isotropic gaussian ($\mu = 0, \sigma = 0.02$), Constant(0) | | | | | |

Table 1: CelebA-10 hyperparameters. We train a simple autoencoder architecture with minimal hyperparameter tuning.

**Degradations.**    The images are scaled to values $[-1, 1]$

- **Noise:** We add Gaussian noise $n = \mathcal{N}(0, 0.15)$
- **Grayscale:** We take the mean of each pixel over the channels and assign to each of the three channels as the model expects three channels.
  $g(x) = x.mean(dim = 1, keepdim = True).repeat(1, 3, 1, 1)$.
- **Sketch:** We divide the pixel values by the Gaussian blurred image pixel values with kernel size of 21. The standard deviation is the default w.r.t. kernel size by PyTorch: $\sigma = 0.3 \times ((kernel\_size - 1) \times 0.5 - 1) + 0.8$. we add and subtract 1 to perform the division on positive values.
  $s(x) = (g(x + 1)/(gaussian\_blur(g(x + 1), 21)) + 10^{-10}) - 1$.

# D    UNCURATED VISUAL RESULTS

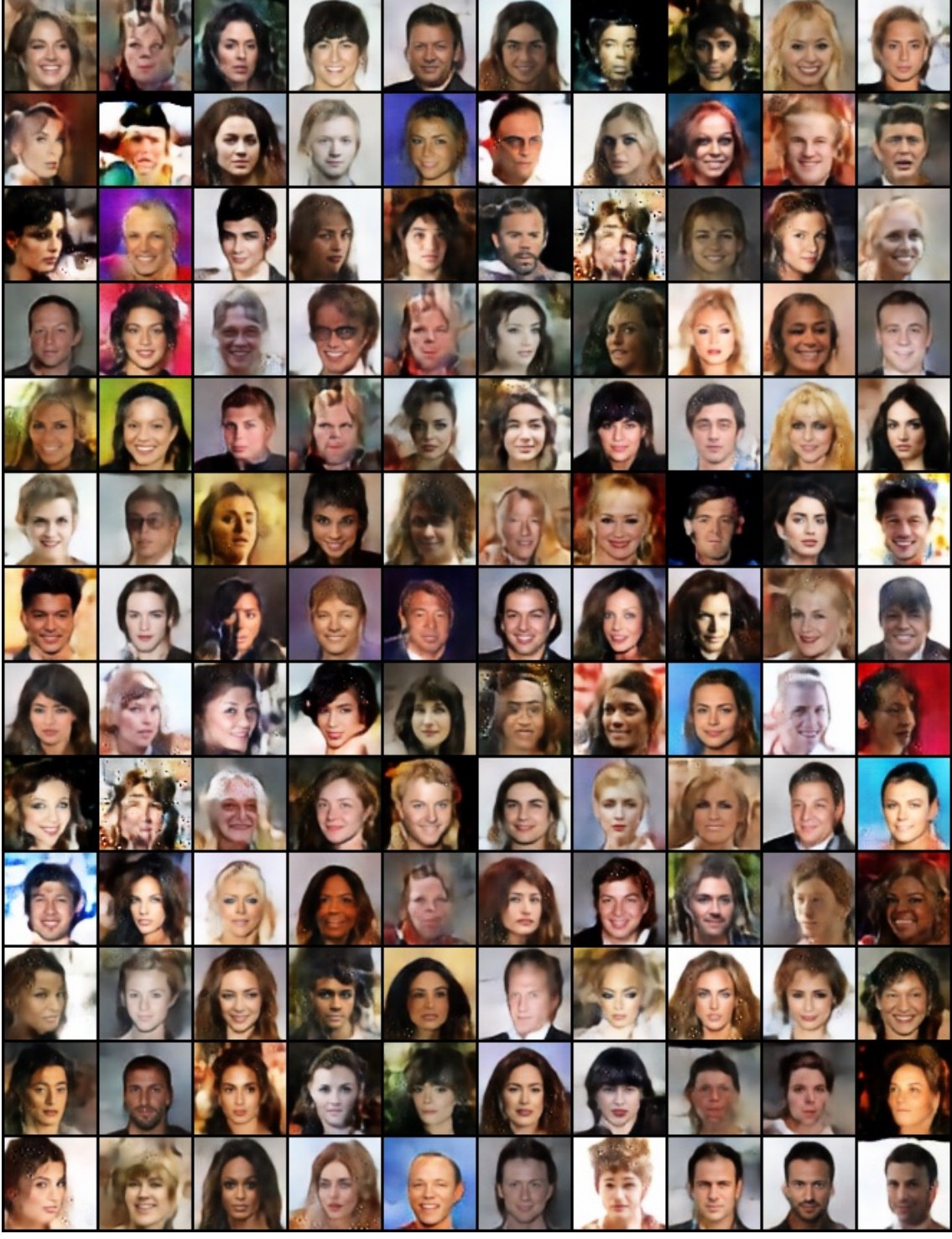

Figure 11: Uncurated CelebA samples from applying IGN once: $f(z)$.

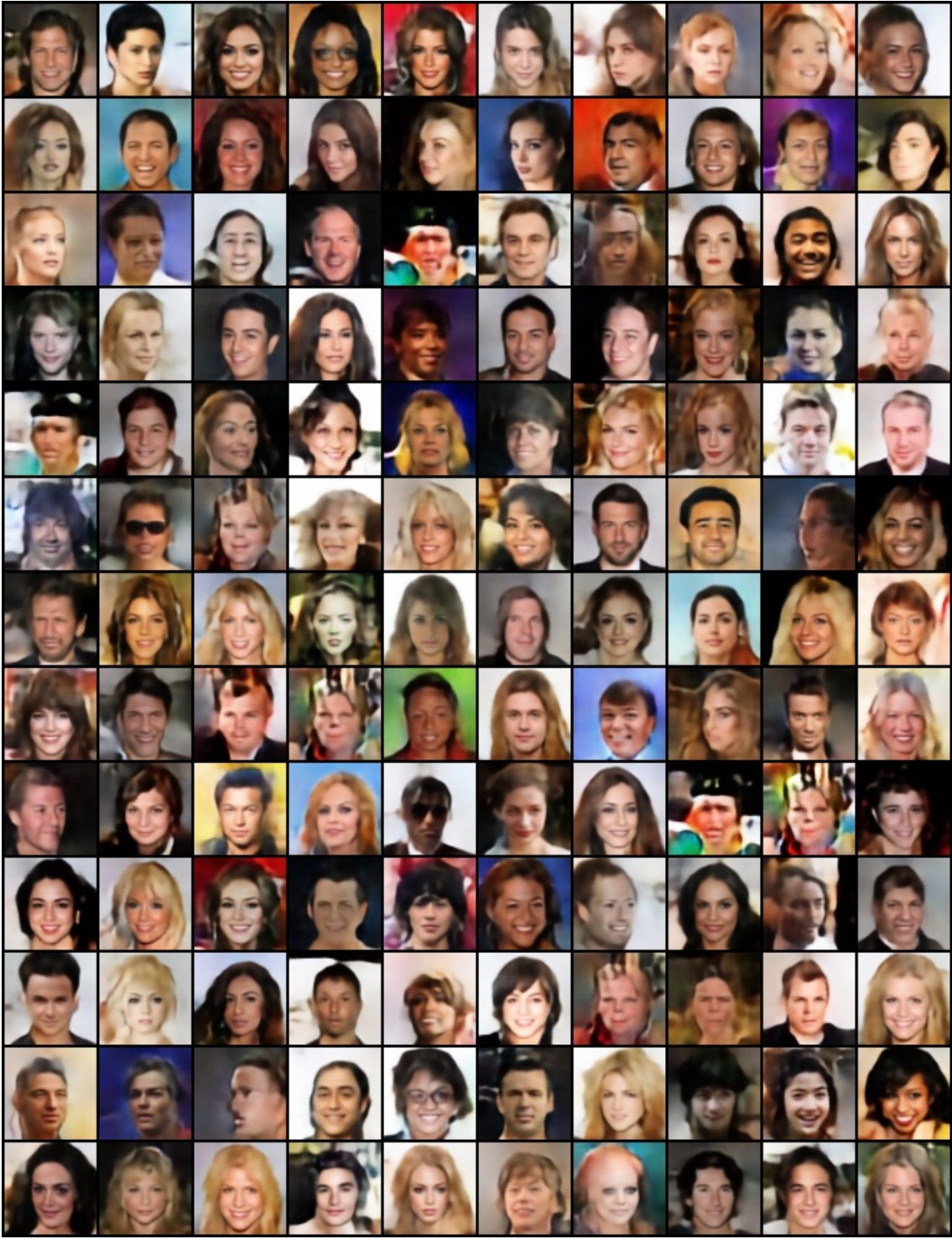

Figure 12: Uncurated CelebA samples from applying IGN twice: $f(f(z))$.

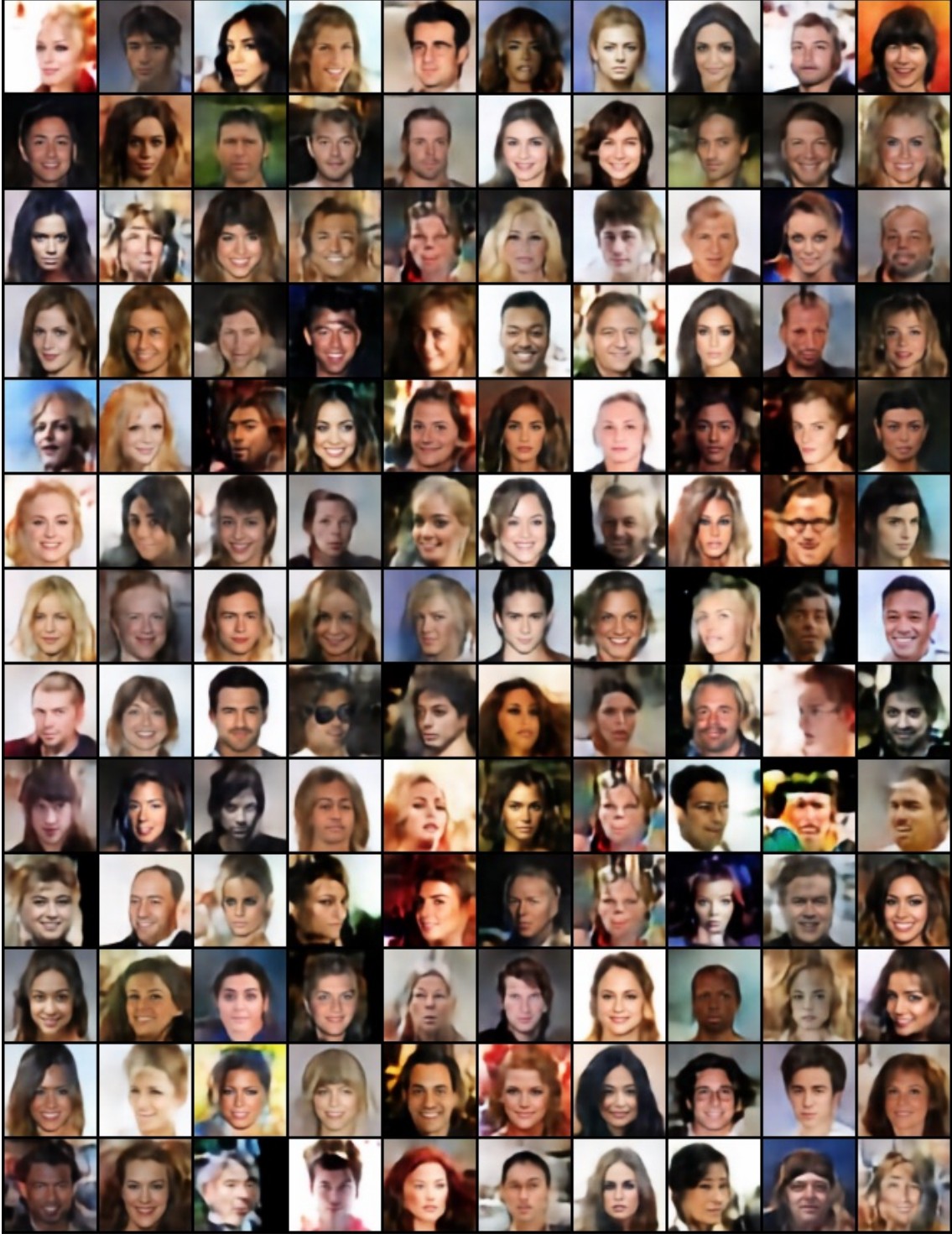

Figure 13: Uncurated CelebA samples from applying IGN three times: $f(f(f(z)))$.

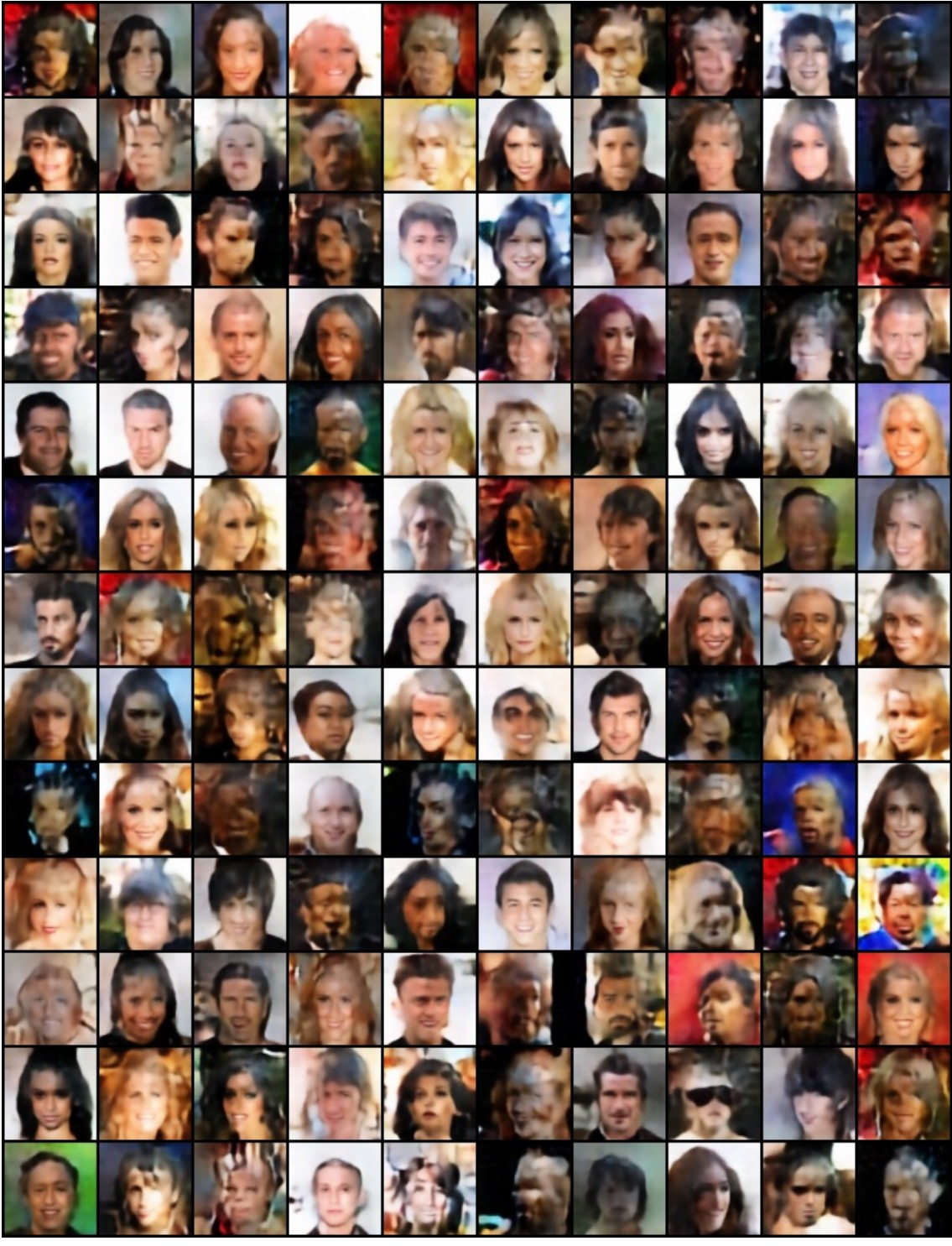

Figure 14: Uncurated CelebA samples from applying IGN four times: $f(f(f(f(z))))$.

## Regular DCGAN Architecture

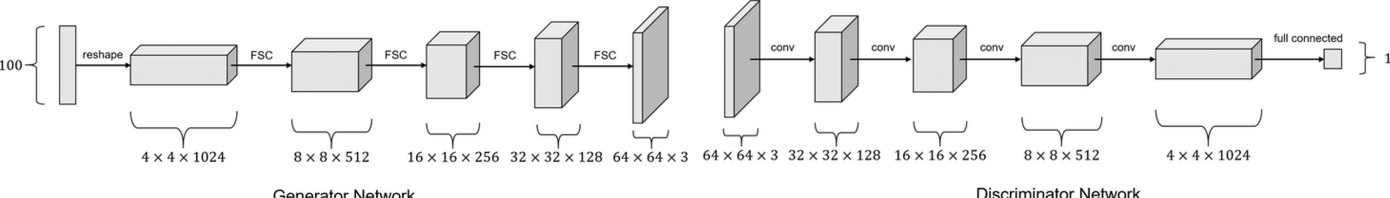

## Flipped DCGAN Architecture, used in IGN for CelebA

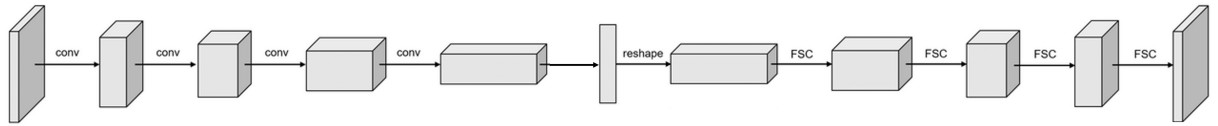

Figure 15: Illustration of the architecture used for CelebA generation. Top shows original DCGAN architecture. Bottom shows that we chop the binary decision head of the discriminator and flip the order to create an hourglass-like architecture. The exact sizes of all the layers are in table 1

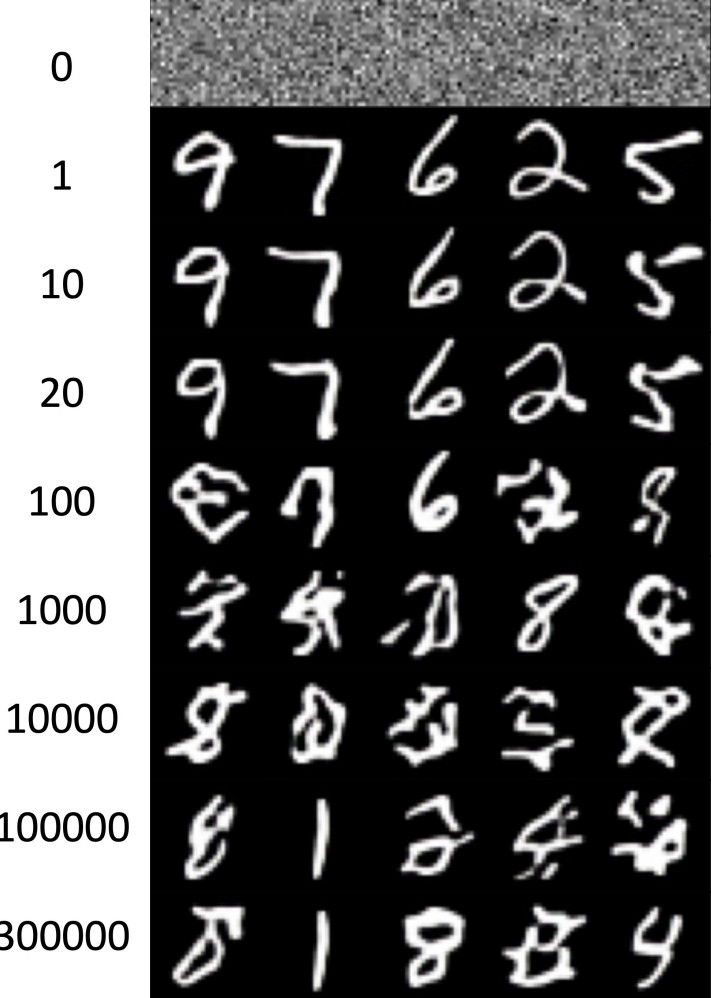

Figure 16: Stress testing by extreme sequential application of IGN. We check what happens for $f^k(z)$ for $k \to \infty$. The numbers on the left indicate $k$, the number of sequential applications. For the first few tens of iterations, the system seems to be stable but around 100 it diverges from the result of the first application. We hypothesize that the size of the model which influences the ability to get low reconstruction loss determines the stability.

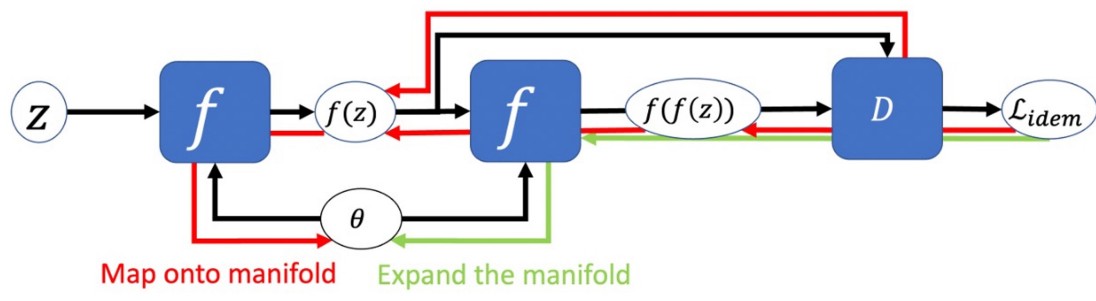

Figure 17: An illustration supporting Fig. 2. **Top:** Naive implementation using $\mathcal{L}_{idem}$ without $\mathcal{L}_{tight}$. Showing two paths of gradient-based optimization of $\theta$, the parameters of $f$. The update for $\theta$ simply consists of a sum of two gradients. The red gradient optimizes $\theta$ s.t. $f$ maps better to the current estimated manifold. The green gradient optimizes so that for a given $f(z)$ the manifold will expand to contain it. **Bottom:** Illustration of the gradients in IGN. As explained in 2.2, The idempotence term is split so that we can negate the the term that expands the estimated manifold in the top illustrations, and therefore tighten it.

