# OpenReview forum: "Idempotent Generative Network"
_ICLR.cc/2024/Conference — ICLR 2024 poster_

### Official Review · Reviewer_j2FZ · 2023-10-27

**Soundness:** 2 fair
**Presentation:** 3 good
**Contribution:** 3 good
**Rating:** 8
**Confidence:** 3

**Summary:**

This work proposes a novel approach to generative modeling based on the idea of idempotence. An idempotent function $f: X \rightarrow X$, is one for which $f(f(x)) = f(x)$ for any $x \in X$. The aim of the model proposed in the paper, Idempotent Generative Network (IGN), is to achieve idempotence so that for any $x$ drawn from the source distribution, $f(f(x)) = f(x)$. The paper provides several justifications for why this property is desirable in the context of generative modeling that essentially amount to IGN having many of the nice properties held individually by either GANs, diffusion models, etc. The paper has an interesting discussion of the loss function needed to train such a model and provides one theoretical result justifying the optimization approach. Several experiments are run, including use of the idempotent property of IGN to project noisy or corrupted examples back to the target distribution.

**Strengths:**

- **Underlying idea:** The use of idempotence in deep learning is appealing given its deep connection to mathematics and theoretical computer science. Because deep learning is essentially a science of function composition, connecting it to category theory and related domains focused on composition in pure mathematics is a rich area for exploration. This is a great example of what first steps in that direction look like. Since idempotents have been studied in great depth, there is likely many additional constructions that could be explored using this work as a starting point.
- **Simplicity of construction and clarity of writing:** IGN is described very clearly so that readers from a range of backgrounds can understand the constructions. The different parts of the model (particularly the loss function terms) are well-justified both mathematically and informally in the text. As might be expected when researchers utilize a truly foundational idea, constructions are surprisingly simple (as was the case in this paper).
- **Quality of the outputs:** Despite the fact that this is a novel generative framework, this reviewer felt that the outputs looked quite good. It was especially exciting to see that that $f(f(f(x)))$ looked mostly the same as $f(x)$. As described below, the reviewer would be interested to see $f^k(x)$ as $x \rightarrow \infty$.
- **Opening paragraph:** Though it is perhaps a small thing, this reviewer really appreciated the opening quote. It captures the underlying idea of the paper and made me laugh.

**Weaknesses:**

- **Motivation:** While the idea is interesting and utilizes foundational structures from mathematics, the motivation for idempotence in generative models is still a little weak. Several different consequences are described in the introduction (IGN provides outputs in a single step, yet allows additional refinements, etc.) Each of these properties is attractive, but already possessed by an existing mature method (as is pointed out in the paper). It was unclear to this reviewer whether starting from scratch with a completely new paradigm is the right way to obtain the desirable properties of GANs, diffusion models, etc. On the other hand, idempotence is a deep idea whose consequences have been explored in a wide range of settings and directions. It feels likely to this reviewer that there are good reasons for wanting idempotence in a model that go beyond properties we already have in alternative methods. However, these may yet need to be uncovered.
- **Comparison to other methods:** While the reviewer agrees that it is not fair to hold this novel approach to the standards of more mature methods, it would still be useful to be able to compare the performance of IGN with known Approach. As it is, this reviewer was unable to tell how far IGN is from the performance of a comparably sized GAN, VAE, or diffusion model.
- **Quantitative results:** It appears that the only way the reader can evaluate the experiments is through a handful of example generations. While these are helpful for getting a qualitative sense of the model (and look good, as described in the Strengths section), it would have made the analysis stronger if quantitative results were also presented. It would be interesting, for instance, to understand network performance for different types of initial $x$, especially since the domain of the network includes both $\mathcal{P}_z$ and $\mathcal{P}_x$. It would have also been interesting to quantitatively measure differences in $f^{(k)}(x)$ as $k \rightarrow \infty$.

### Nitpicks

- Idempotence is spelled wrong in the sentence below equation (7).
- The tightness objective is a little subtle. It might make sense to utilize some of the notation already introduced (e.g., $\mathcal{S}$) to help communicate why this loss term is necessary.
- It may be a matter of personal taste, but this reviewer felt that while the PyTorch code was appreciated, it might be better to include in the Appendix and instead include additional analysis of the experiments.

**Questions:**

- One of the claimed strengths of IGN is that through the idempotence property, it can be used to project noisy or corrupted images back to the data manifold. Based on the reviewer’s understanding of the capability, there are already a range of methods for doing this (e.g., via image denoising with a diffusion model). Is there some additional nuance that the reviewer is missing? It is certainly the case that existing methods will not map an image plus noise $x + \epsilon$ by to $x$ exactly, but it was our sense from reading the paper that neither would IGN.
- It would be interesting to see the behavior of $f^{k}(x)$ as $k \rightarrow \infty$, does this converge or diverge? If it does converge, what does it converge to?
- This reviewer did not understand the modified GAN network architecture described in Section 2.3.
- As the authors note, the development of high-performing generative models has come about through numerous iterative improvements. Would it be possible to build an IGN model using the framework of a known generative approach?
- What are the current drawbacks to scaling this method up?

---

> ### Author Response · Authors · 2023-11-13
> **Thank you for the thoughtful consideration of the paper and constructive feedback.**
>
> **Motivation: “Several different consequences are described in the introduction (IGN provides outputs in a single step, yet allows additional refinements, etc.) Each of these properties is attractive, but already possessed by an existing mature method (as is pointed out in the paper). “**
> IGN has some unique properties that do not exist in other models. In addition, IGN has some properties that exist in other models but do not co-exist together. For instance, let us consider the single step + refinement property the reviewer mentions. There are Diffusion based methods with fewer steps, such as consistency models. However IGN always does one step full generation after which it is possible to apply it again for refinement.
> Another example is a consistent latent space. Surely, GANs have that, but IGN allows smooth interpolation across all iterations ($f(z), f(f(z)), f(f(f(z)))$...). (See video in supplementary).
> Also, Unlike GANs and Diffusion models, editing of real images in IGN does not require any type of inversion. Simply inputting a real image in a feed-forward fashion into IGN allows manipulation on it (see new figures 9,10 in revised manuscript).
> Lastly there is the projection property addressed below in response to your specific question.
>
> **“IGN can be used to project corrupted images … already a range of methods for doing this (e.g., via image denoising with a diffusion model).”**
> While many methods handle certain degradations or modifications of images, they will typically be trained for it specifically (e.g., Diffusion-Denoising, Cold Diffusion, image enhancement methods, image-to-image methods), or they will require some manipulation at test time (adding noise, guidance in diffusion and some tricks in GANs). IGN is not trained for specifically on modified images. IGN only sees pure noise and real images during training. No different type of inference is needed. For example, no noise is added to bring the input into distribution, as would be in, for instance, SDEdit. Note that it is not only for corrupted images but also image-to-image translation such as sketches. It can be thought of as a generic  ``make-it-real" button.
>
> **“ $\mathbf{f^k(z) \quad k\rightarrow\infty}$. Does this converge or diverge?“:**
> We added fig.16 in the appendix of the revised manuscript which demonstrates applications of MNIST trained IGN up to $300000$ iterations. For the first few tens of iterations, the system seems to be stable but around 100 it diverges from the result of the first application. We hypothesize that the size of the model which influences the ability to get low reconstruction loss determines the stability.
>
> **“This reviewer did not understand the modified GAN network architecture described in Section 2.3.”**
> In the revised manuscript, we added Fig.15 to illustrate the modification of DCGAN architecture to be used for IGN on CelebA. The exact architecture is in Table.1 in the appendix.
> DCGAN has a generator $G$ and a discriminator $D$. In training $D(G(z))$ is applied. $D$ produces a $1\times1\times1$ result per image. we modify $D$ to end with a $512\times1\times1$ tensor instead, denoted as $\tilde{D}$. Our model in terms of DCGAN is $G(\tilde{D}(z))$.
>
> **Tightness loss clarity:**
> Indeed it is subtle, and challenging to convey. We added the notion of $\mathcal{S}$ to the text around fig.2 and its caption as you suggested. We updated the text about $\mathcal{L}\_{tight}$ and also added Fig.17 that offers further clarification on the two paths to enforce idempotence. We welcome any suggestions to enhance the explanation in the paper.
> We have provided some informal perspectives to help understand $\mathcal{L}\_{tight}$ in responses to other reviewers. please check them (not pasting here due to char limit, sorry.).
>
> **“Would it be possible to build an IGN model using the framework of a known generative approach?”, “What are the current drawbacks to scaling this method up?
> “**
> Scaling up is a current effort. Indeed, the ways explored are based also on existing methods. Among others, working in latent space as in Stable Diffusion and using transformer architectures are being tried. One challenge we have seen is the need of a bottleneck which makes the popular Unet challenging to work with due to its skip connections.
>
> **Quantitative evaluation and comparison to other methods:**
> We report FID=39 for IGN on CelebA. Naturally, this is not competitive with modern models, but is comparable to early models such as DCGAN (FID=34), while also capable of a wide range of conditional tasks. We updated the draft paper to include this info.
>
> **Idempotence spelled wrong:** Fixed, thanks!
>
> **Code better in appendix:**
> The provided PyTorch code is a substitute for an algorithm pseudocode. We felt that it is the cleanest way of presenting the algorithm. That said, we can consider moving it to the appendix and replace it with the regular pseudocode notation. We will also release full code.

---

> ### Author Response · Authors · 2023-11-20
> **Followup**
>
> Dear reviewer, towards the end of the discussion phase, we trust that our response has successfully addressed your inquiries. We look forward to receiving your feedback regarding whether our reply sufficiently resolves any concerns you may have, or if further clarification is needed.

---

> > ### Comment · Reviewer_j2FZ · 2023-11-21
> > **Thanks for clarification and updates**
> >
> > The reviewer would like to thank the authors for their clarifications and updates.
> > - While the reviewer now has a better sense of the strengths of the IGN approach, they still feel that the motivation for a new approach needs to be sufficiently compelling to overcome the advantages of existing finely polished methods. In this light, the reviewer feels that the motivation is still somewhat underwhelming. That being said, the field would be unhealthy if new (or perhaps not so new if one believes the public comment above) methods were not routinely explored.
> > - The reviewer enjoyed the experiments investigating what happens to $f^k(x)$ as $k \rightarrow \infty$. It is not clear to the reviewer in what ways the size of the model would impact stability, but it is a direction that would be interesting to explore in the future.
> >
> > Overall, the reviewer feels that the work is valuable because of the way that it frames the generative task around the idempotence property, the interesting analysis of the task, and the fact that the model performs at around the same level as other (now successful) approaches at similar points in their development. The reviewer would be happy to raise their score to an 8.

---

> > > ### Author Response · Authors · 2023-11-21
> > >
> > > Thank you for the time and effort you have dedicated to reviewing our paper. Your thorough review and insightful observations have significantly contributed to improving the quality of our work.

---

### Official Review · Reviewer_5afM · 2023-10-31

**Soundness:** 3 good
**Presentation:** 3 good
**Contribution:** 3 good
**Rating:** 5
**Confidence:** 3

**Summary:**

This paper introduces a new class of generative model that is based on idempotent, i.e., $f(f(x)) = f(x)$. It acts like a projection function that project anything to the target data manifold, and it preserves identity for data already in the target manifold.To train such a model, first a reconstruction loss is used to let the model map latent z to the data manifold, and an idempotent loss to ensure the property of idempotent. An additional tightness loss is used for regularization. Some preliminary results of generation is shown.

**Strengths:**

1. The biggest strength is that the idea is novel and refreshing. It introduces a new class of generative models with an interesting formulation.

2. Overall the presentation is good. The idea is clearly stated in the pseudo code. The introduction of idempotent is clear and interesting in the first section.

3. Although the results are preliminary, there are some interesting observations. For example, the method can naturally take things other than noise as input (e.g., Figure 5 c)

**Weaknesses:**

1. The biggest issue is that I don't get the motivation for designing such a model. When one designs a new class of generative model that is different from any previous approaches, it is necessary to clarify that it has the potential to address some limitations of previous approach. As a preliminary study, it does not need to have great results, but conceptually the advantage of the model has to be stated. Otherwise, it is just "another class of generative models". Since it suffers the same mode collapse and instability issue of GAN, the motivation becomes more important. Why do we need such a model? What are the potential advantages?

2. I don't quite get the point of the tightness loss. Why maximize it is good for regularization? More explanation is needed.

**Questions:**

My questions are stated in the weakness section.

---

> ### Author Response · Authors · 2023-11-13
>
> Thank you for the thoughtful consideration of the paper and constructive feedback.
>
> **Motivation: “When one designs a new class of generative model that is different from any previous approaches, it is necessary to clarify that it has the potential to address some limitations of previous approach.”**
> IGN has several potential advantages over the current methods:
> 1. Unlike DIffusion models, IGN generates in a single step, and then, unlike GANs,  allows for sequential refinement. There are Diffusion based methods with fewer steps, but for IGN the number of reprojections can be determined adaptively (apply again if desired). A single step will always yield good results.
> 2. Unlike Diffusion models, IGN maintains a consistent latent space, allowing smooth interpolation across all iterations ($f(z), f(f(z)), f(f(f(z)))$...). (Please see videos in supplementary).
> 3. IGN can be applied to types of data that it has never ‘seen’ and project them onto the data manifold. While Diffusion models can be guided to get condition signals, IGN does that when regularly applied to degraded/modified input.
> 4. Unlike GANs and Diffusion models, editing of real images does not require any type of inversion. Simply inputting a real image in a feed-forward fashion into IGN allows manipulation on it (see new figures 9,10 in revised manuscript).
>
>
> **”I don't quite get the point of the tightness loss. Why maximize it is good for regularization? More explanation is needed.”**
> We appreciate the feedback and acknowledge the challenge in conveying the concept of $\mathcal{L}\_{tight}$. In the new draft, we added the notion of $\mathcal{S}$ to the caption of fig.2. We updated the text about $\mathcal{L}\_{tight}$ and also added Fig.17 to the appendix of the revised manuscript. Fig.17 offers further clarification on the two paths to enforce idempotence. We offer additional informal perspectives here for better clarity and welcome any suggestions to enhance the explanation in the paper.
>
> Consider the metaphor of High Jump. The objective is to clear the bar. This can be achieved in two ways: jumping higher, or lowering the bar. If the athlete, $f$ trained directly for this objective, it would become an expert on lowering the bar (tends to become identity). Instead, the athlete tries to get **bigger** gap from the bar when jumping, but trying to get **smaller** gap from the bar (opposite from the objective), when setting it according to its recent jumps. We emphasize that this is the same athlete that both jumps and sets the bar.
>
> Examining the objective $\delta\_\theta = D(f\_\theta(f\_\theta(z)),f\_\theta(z))$, we can split it into two phases:
> (1) $y=f\_\theta(z)$ (jumping)
> (2) $\delta\_\theta = D(f\_\theta(y),y)$ (evaluating the gap from the bar)
> Both are done using the same set of parameters $\theta$. See Fig.17 in the appendix. The instance of $f$ that is in phase (1) sends gradients to $\theta$ that improve the generation (higher jump, lower drift $\delta\_\theta(y) = D(f\_\theta(y), y)$). The Gradients to $\theta$ through the instance of $f$ that is in phase (2) judge generated images and train to exclude them from the manifold (higher drift), just like setting the bar higher according to the recent jumps.
> This aspect can also be thought of as a GAN where the generator and discriminator are the same network.

---

> ### Author Response · Authors · 2023-11-20
> **Followup**
>
> Dear reviewer, towards the end of the discussion phase, we trust that our response has successfully addressed your inquiries. We look forward to receiving your feedback regarding whether our reply sufficiently resolves any concerns you may have, or if further clarification is needed.

---

> > ### Comment · Reviewer_5afM · 2023-11-23
> > **Thanks for the response**
> >
> > I would like to thank the authors for the response. My confusion on the tightness loss is addressed. However, I don't quite buy the motivation part. Yes, it is a single step method compared to diffusion, but so does GAN. If you claim advantages like sequential refinement, it becomes multi-step again. In addition, there isn't enough evidence showing that the refinement indeed significantly improve the sample quality. Overall, the minimax loss objective makes it look like a GAN variant, which is also pointed out by someone else. It suffers from the same issue of mode collapse of GAN, and the fact that it has only a single network does not distinguish itself enough. EBGAN is one such example, and the well-known connection between maximum likelihood training of EBM and adversarial training is another, where there's only one network for discrimination, and the generator is instantiate by the iterative sampling.
> >
> > As a result, given the connection to earlier models and the behaviors exhibits, I think the proposed model does not look very promising. It's a good idea to shed new lights and find new connections to existing frameworks, but that would require a complete rewrite. Therefore, I maintain my judgement on the submission.

---

> > > ### Author Response · Authors · 2023-11-23
> > >
> > > Thank you for your comments and insights.
> > > We are happy that we were able to clarify $\mathcal{L}\_{tight}$ and thank you for acknowledging it. We will address the specific comments you made, trying to alleviate the concerns, as we believe there is a fitting explanation for most of them:
> > >
> > > - "... sequential refinement, it becomes multi-step again":
> > > We want to highlight the difference. In IGN you perform one good step. Then you can choose to reapply $f$ again. In GANs you will have this one step. In diffusion models you generate noisy images over predetermined steps, only the last one is clean. So IGN introduces the possibility for one big step, followed by possible much smaller steps.
> > >
> > > - "there isn't enough evidence showing that the refinement indeed significantly improve the sample quality":
> > >  In our results $f(f(z))$ is consistently of better quality than $f(z)$. Please see fig.7 and fig.8 in our appendix. Especially the MNIST digits 4,5,7,8 and the rightmost celebA image. the figures show how the quality improves over 3 steps $z$->$f(z)$->$f(f(z))$->$f(f(f(z)))$.
> > >
> > > - "minimax loss objective":
> > > IGN has no minimax loss objective, it cannot be formulated as such since there is only one model- one set of parameters. This is the reason for the theoretical derivation of convergence to target distribution that is different from the derivation with GANs.
> > >
> > > - Relation to EBMs, "where there's only one network for discrimination, and the generator is instantiate by the iterative sampling":
> > >  This relation is acknowledged in the paper, the equivalence and difference are reviewed: "IGN introduces a similar, yet distinct paradigm: rather than $f$ acting as the energy function, this role is filled by $ \delta(y) = D(f(y), y) $. The model trains $f$ to be idempotent, with the objective to minimize $\delta(f(z))$. A successful training procedure would align the range of $f$ with the low-energy regions as measured by $\delta$. This reduces the need for separate optimization procedures to find the energy minimum."
> > > So IGN can be viewed as an EBM, where there is no need for iterative sampling, because the function $f$ is already mapping to low energy.
> > > Specifically for EBGAN, we will now write a thorough response to the public comment.
> > >
> > >
> > > We appreciate the response and the discussion and hope we provided information that is helpful to clarifying the points made.

---

### Official Review · Reviewer_Cz9y · 2023-11-01

**Soundness:** 2 fair
**Presentation:** 3 good
**Contribution:** 2 fair
**Rating:** 3
**Confidence:** 4

**Summary:**

This paper presents a new kind of generative model for image synthesis called the Idempotent Generative Network (IGN). The key idea is to learn a function $f$ such that $f(x) = x$ if $x$ is on the data manifold, and $f(z)$ is on the data manifold if $z \sim P_0$ for some prior $P_0$ (in which case $f(f(z)) = f(z)$). The network $f$ is learned with three loss terms: a reconstruction loss to encourage data samples to be mapped to themselves, an idempotent loss to encourage two applications of $f$ to a state $z$ from the prior to match a single application of $f$, and a tightness objective encouraging $f$ to map any state $z'$ not on the manifold as far away from the manifold as possible. A theoretical analysis is presented which claims that the proposed method will generated samples from the data distribution if $f$ is perfectly trained and has enough capacity. Experimental results show that the proposed method can generate MNIST and CelebA images from noise.

**Strengths:**

* The work is an ambitious and refreshing approach to generative modeling. Idempotence is an interesting foundation for building a generative model. Overall the paper was enjoyable to read.
* The central ideas are clearly presented and I could easily follow the training algorithm and motivation of the method.
* Empirical results show that the method is capable of image generation and editing the features of generated faces.

**Weaknesses:**

* I am not fully convinced of the theoretical validity of the method. In particular, Equation (19) does not appear fully rigorous. From my understanding, the value $M$ is the maximum distance between points in the state space.
  * What if the state space is unbounded? Should we expect states to be restricted to an image hypercube?
  * Why is it the case that $max_{\theta*} \delta_{\theta*}(y) = M$ for any $y$ when $P_x (y) < \lambda_t P_{\theta*} (y)$? Wouldn't that mean the distance between $y$ and the furthest point from $y$ in the state space is $M$ for each $y$? I am not sure that makes intuitive sense to me. It seems that $M$ should not be a constant but rather a function $M(y)$ measuring the distance between $y$ and the furthest point in the state space, which greatly complicates the analysis that follows.

  The proof heavily relies on the simple form of (19), but I am not sure this simple form is valid. Furthermore, in practice the distance $M$ is heavily restricted which leaves a gap between the theoretical development of the method and practical implementation. Since this method departs heavily from the established probabilistic principles used to justify generative models, clearly demonstrating the validity of the method is essential. I am willing to raise my score if these points can be addressed.
* Although I understand that the current results are meant to be a proof of concept, it would still be good to provide a more formal comparison with prior generative models e.g. by calculating FID scores on CelebA.

**Questions:**

Please see my questions about the theoretical justification of the method in the weaknesses section.

---

> ### Author Response · Authors · 2023-11-13
>
> Thank you for the thoughtful consideration of the paper and constructive feedback.
>
> **The value $M$:**
> $M$ is the supremum of the distance metric for any pair $(y\_1, y\_2)$. There is no bounded space assumption -- M can also be infinity (which is why we did not define it as Maximum. We added a clarification in the paper). We examine both the scenario of unbounded $M$ (which is the case of MSE as used for MNIST and shown in the provided code), and the scenario of bounded $M$ which exists when using the clamp technique as introduced in eq.14.
> Unbounded $M$ is consistent with all the steps of the proof, under the practical assumption that $M \cdot 0 < M$.
>
> **The role of $M$ in eq.21 (formerly eq.19):**
> Thank you for pointing out the lack of clarity. We have revised the text before the equation (that is eq.21 now).
> Eq.21 describes what happens when $\mathcal{L}\_{rt}$ is minimized, according to how it is expressed in eq.20 (eq.18 second row in pre-revised version):
> $$ \mathcal{L}\_{rt}(\theta;\theta^*)=\int \delta\_{\theta}(y) \Bigl(
> \mathcal{P}\_x(y)
> \- \lambda\_t
> \mathcal{P}\_{\theta^*}(y)
> \Bigr)dy $$
> $\delta\_\theta$ is multiplied by a given expression in the brackets that is not being optimized in this phase.
> To obtain a minimum, $\delta\_\theta\geq0$ needs to be as big as possible when the expression in the brackets is negative and as small as possible when positive. Thus, for any $M>0$ eq.21 holds when  $\mathcal{L}\_{rt}$ obtains its minimum:
> $$ \delta\_{\theta^*}(y) = M \cdot 1\_{\{\mathcal{P}\_{x}(y) < \lambda\_t  \mathcal{P}\_{\theta^*}(y) \}}$$
>
> **Should it be $M(y)$?:**
> Thank you for identifying this subtle issue. Indeed, for a general metric in a general space, $M$ should be defined using a reference input $M(y\_1) = \sup\_{y\_2} D(y\_1, y\_2)$. An example could be MSE in some bounded space. We should point out that in the metrics used in the paper (MSE, L1, and clamped L1), $M$ would be constant ($\infty$ for unbounded metrics and the clamp value for the clamped metric). Regardless, using $M(y)$ instead of $M$ would not have influence on the subsequent steps of the proof as long as $M(y)>0\quad\forall y$.
> To see this we can examine minimizing $\mathcal{L}\_{idem}$ as in eq.24 modified with $M(y)$:
> $$
>     \theta^* = argmin\_\theta  \mathbb{E}\_{z}\bigl[ M(y) \cdot  1\_{\{
>     \mathcal{P}\_{x}(y) < \lambda\_t
>     \mathcal{P}\_{\theta}(y)
>     \}} \bigr]
> $$
> $M$ being positive implies that for every value of $y$, $M(y)\cdot 0 < M(y) \cdot 1$. This leads to the same $\theta^*$ for constant $M$ or $M(y)$.
> We are happy to revise and redefine $M(y)$. Another option is to add in the text assumptions on the metrics used.
>
> **$M$ imposes gap between theory and practice?:**
> Hopefully, the above explanation was able to show that $M$ does not impose restrictions and that, while the theoretical results make very strong idealistic assumptions, the practical implementation does not diverge more than in any other generative model's theoretical results.  But please let us know if something is still unclear and we will do our best to respond quickly.
>
>
> **FID on CelebA:**
> We report FID=39 for IGN on CelebA with the architecture in Table.1. Naturally, this is not competitive with modern models, but is comparable to early models such as DCGAN (FID=34), while also capable of a wide range of conditional tasks. We updated the draft paper to include this info.

---

> ### Author Response · Authors · 2023-11-20
> **Followup**
>
> Dear reviewer, towards the end of the discussion phase, we trust that our response has successfully addressed your inquiries. We look forward to receiving your feedback regarding whether our reply sufficiently resolves any concerns you may have, or if further clarification is needed.

---

> ### Comment · Reviewer_Cz9y · 2023-11-22
> **Revision including EBGAN would be helpful**
>
> I read the response of the author, other reviewers, and the public commenter. I agree with the public commenter that this method is a special case of EBGAN where the generator and discriminator share the same structure. As noted by the author response, there are interesting properties of an idempotent network that EBGAN does not have and that the connection between idempotence and EBGAN is an interesting novel view that is perhaps more streamlined than the original EBGAN framework. I suggest that the authors simply use the EBGAN proof of convergence to theoretically justify their method. Overall, I believe this paper needs a major revision to incorporate EBGAN as a prior work and reframe its contributions. I will keep my score.

---

> > ### Author Response · Authors · 2023-11-22
> >
> > Thank you for your feedback.
> > We would like to emphasize that in IGN the 'generator' and 'discriminator' do not merely "share the same structure", but are in fact the same network with the same parameters-- the same entity being optimized. Moreover, the IGN convergence cannot be proved by any GAN, including EBGAN proof, as it simply cannot be formalized as a two player game.  The difference is also in the training regime: IGN does not alternate between two networks competing with each other.  Even if we put all of these a side, and claim that IGN is a GAN where the generator and discriminator are the same model with the same parameters-- We consider this a non-trivial leap, and the main point of our work. The notion of "self-adversarialness" we introduced underscores this uniqueness.
> >
> > We acknowledged the relation to both GANs and EBMs in the paper, thoroughly describing the analogies and we will definitely add a thorough discussion specifically on EBGAN as well. However, we point out that IGN is
> > - Different in nature (same $f$ for generator and critic),
> > - Different in its theoretical framework (convergence can't be shown the same way as in GANs/EBGAN due to the shared parameters, a different approach is needed),
> > - Different training regime (no alternating, one set of parameters accumulates gradients from all losses),
> > - Has many important capabilities that GANs/EBGAN do not have (acknowledged by the reviewer, listed in the response to the public comment).
> > - Associates idempotence to generative modeling, a connection that to the best of our knowledge was not shown before, including in GANs or EBGAN.
> >
> > We would like to also make sure that the concerns about our proof and the value $M$ have been clarified and that you do not have further questions regarding that.

---

> > > ### Author Response · Authors · 2023-11-22
> > > **Manuscript revised**
> > >
> > > We would like to thank you once again and inform you that we have revised the manuscript to include a detailed paragraph on EBGAN, stating that "EBGAN is the closest existing model to IGN", and then discussing the similarities and differences. We plan to add more revisions, stemming from the discussion with you, but we are submitting this current draft now so that we can hopefully get more feedback before discussion period ends.
> > >
> > > An additional point we've included in the revised manuscript, and would like to share here, concerns the EBGAN discriminator: Differently from IGN, "the EBGAN discriminator is not trained to project as it only sees real or generated images, but not latents or out-of-distribution instances. It is trained to discriminate.". We highlight this because it shows EBGAN misses a key property of IGN, where $f$ doesn't only judge the quality, it also tries to improve it. Idempotent models are unique in the sense of acting like Autoencoders for realistic data, similarly to the EBGAN discriminator, but as conditional generators for other inputs.
> > >
> > > Hope we could clarify both the concern about EBGAN and the theoretic derivation, please let us know if any further clarification needed.

---

### Official Review · Reviewer_Sz9m · 2023-11-01

**Soundness:** 4 excellent
**Presentation:** 3 good
**Contribution:** 4 excellent
**Rating:** 8
**Confidence:** 4

**Summary:**

This work presents what appears to me to be a novel way to model the generative process from noise (or corrupted images) using idempotence as the underlying principle for the objective(s). Notably, they train a model which at optimum should maintain given datapoints from the target distribution, yet generate samples whose manifold is maximally "tight". These objectives together yield a novel generative model algorithm, and there is theoretical and empirical support.

--- Update post rebuttal / discussion

I've looked over the discussion of EBGAN vs IGN, as well as revisited the EBGAN paper and reread the submission. While the public commenter's insights are useful, I feel that the reviewers have done a good job addressing them: the motivation of IGN is certainly different from those of EBGAN's, and the story is still clear and easy to follow. I'm pleased we've found a way to think about training generative models that leads to an algorithm that is very similar to a prior work, but I do see the author's points about the differences, both in terms of how the model is optimized and how it is evaluated, as being well aligned with the overall story of the paper. The added discussion wrt EBGAN also is important and sufficient.

I'm less pleased with the intention of the public reviewer to bring the story under alignment EBGAN. It's true there are many open questions from the discussion, but it isn't necessarily the job of this paper to do so. It's an interesting motivation for generative modeling that bears fruit in something that is close, but not precisely the same as (nor subsumed by) prior works. If the public commenter wants to delve into this problem more, I'd encourage them to do so, rather than making that a criteria for acceptance to this conference.

Moreover, EBGAN is in no way harmed by how the paper is presented. No incorrect claims are made that I can find, and the motivation (idempotence) *stands on its own*.

**Strengths:**

Overall I found the approach to be appealing and interesting. Though there isn't necessarily a strong compelling reason here why to use idempotence to train a generative model, I don't find this as a weakness (prefacing here), as there is scientific curiosity here, the intuition is compelling, the theory seems correct wrt the optima fitting the target distribution, and the empirical results support everything neatly. I'd also mention that the results are stellar / SOTA, but as noted in the paper: this is fine. Engineering might improve things or it might not, just generally it looks like good science with a solid story and results to support.

**Weaknesses:**

Some points could be clarified, notably I found the discussion around Figure 2 to be less clear than I would have liked. I think I understand the general gist, but I honestly didn't think I understood how the different objectives work together until I got through the theory section. Could this be cleaned up a bit / made a bit more clear? I'd enjoy a bit of some discussion here to see if we can arrive at a better way to present this.

**Questions:**

Why does M:= sup(D(y1, y2)) show up in eq 19? Are we saying if y is outside the target data manifold + the margin that the repeated applications of f should maximize the distance? How is this guaranteed?

---

> ### Author Response · Authors · 2023-11-13
>
> Thank you for the thoughtful consideration of the paper and constructive feedback.
>
> **"I found the discussion around Figure 2 to be less clear than I would have liked."**
> We appreciate the feedback and acknowledge the challenge in conveying the concept of $\mathcal{L}\_{tight}$. We added the notion of $\mathcal{S}$ to the text around fig.2 and its caption. We updated the text about $\mathcal{L}\_{tight}$ and also added Fig.17 to the appendix of the revised manuscript. Fig.17 offers further clarification on the two paths to enforce idempotence. We offer additional informal perspectives here for better clarity and welcome any suggestions to enhance the explanation in the paper.
>
> Consider the metaphor of High Jump. The objective is to clear the bar. This can be achieved in two ways: jumping higher, or lowering the bar. If the athlete, $f$ trained directly for this objective, it would become an expert on lowering the bar (tends to become identity). Instead, the athlete tries to get **bigger** gap from the bar when jumping, but trying to get **smaller** gap from the bar (opposite from the objective), when setting it according to its recent jumps. We emphasize that this is the same athlete that both jumps and sets the bar.
>
> Examining the objective $\delta\_\theta = D(f\_\theta(f\_\theta(z)),f\_\theta(z))$, we can split it into two phases:
> (1) $y=f\_\theta(z)$ (jumping)
> (2) $\delta\_\theta = D(f\_\theta(y),y)$ (evaluating the gap from the bar)
> Both are done using the same set of parameters $\theta$. See Fig.17 in the appendix. The instance of $f$ that is in phase (1) sends gradients to $\theta$ that improve the generation (higher jump, lower drift $\delta\_\theta(y) = D(f\_\theta(y), y)$). The Gradients to $\theta$ through the instance of $f$ that is in phase (2) judge generated images and train to exclude them from the manifold (higher drift), just like setting the bar higher according to the recent jumps.
> This aspect can also be thought of as a GAN where the generator and discriminator are the same network.
>
> **"Why does M:= sup(D(y1, y2)) show up in eq 19?"**
> Thank you for pointing out the lack of clarity. We have revised the text before the equation (that is eq.21 now).
> Eq.21 describes what happens when $\mathcal{L}\_{rt}$ is minimized, according to how it is expressed in eq.20 (eq.18 second row in pre-revised version):
> $$
> \mathcal{L}\_{rt}(\theta;\theta^*)=\int \delta\_{\theta}(y) \Bigl(
> \mathcal{P}\_x(y)
> \- \lambda\_t
> \mathcal{P}\_{\theta^*}(y)
> \Bigr)dy
> $$
>
> $\delta\_\theta$ is multiplied by a given expression in the brackets that is not being optimized in this phase.
> To obtain a minimum, $\delta\_\theta\geq0$ needs to be as big as possible when the expression in the brackets is negative and as small as possible when positive. Thus, for any $M>0$ eq.21 holds when  $\mathcal{L}\_{rt}$ obtains its minimum:
> $$ \delta\_{\theta^*}(y) = M \cdot 1\_{\{\mathcal{P}\_{x}(y) < \lambda\_t  \mathcal{P}\_{\theta^*}(y) \}}$$

---

> ### Author Response · Authors · 2023-11-20
> **Followup**
>
> Dear reviewer, towards the end of the discussion phase, we trust that our response has successfully addressed your inquiries. We look forward to receiving your feedback regarding whether our reply sufficiently resolves any concerns you may have, or if further clarification is needed.

---

### Author Response · Authors · 2023-11-13

We thank the reviewers for their thoughtful comments. We appreciate that all four reviewers highlighted the method's novelty, e.g.  “appealing and interesting” (Sz9m), “ambitious and refreshing approach to generative modeling” (Cz9y), “novel and refreshing” (5afM), “appealing given its deep connection to mathematics and theoretical computer science” (j2FZ). The reviewers also point out the clarity of the presentation: “ideas are clearly presented and I could easily follow” (Cz9y), “idea is clearly stated in the pseudo code. The introduction of idempotent is clear and interesting in the first section.” (5afM), “IGN is described very clearly… different parts are well-justified both mathematically and informally in the text. As might be expected when researchers utilize a truly foundational idea, constructions are surprisingly simple.” (j2FZ).

The main topics the reviewers commented on were the value of $M$ in the theoretical derivation, the explanation of $\mathcal{L}\_{tight}$ and the motivation for a new generative model. We respond to each reviewer individually about these topics and others. We have revised the manuscript to address these comments, and fixed some issues we spotted by ourselves. Main changes are listed below. We have a longer list of all the minor changes too and can post it here if requested.

1. Method: Added the direct gradient path to $f(z)$ in eq.13 and in fig.3. Modified $\partial\delta(f(f(z))$ $\rightarrow$ $\partial\delta(f(z))$.
2. Theoretical Results: Reorganized assumptions and claims in Theorem 1. Defined $M$ close to its first appearance, added clarification that $M$ is unbounded.
5. Experimental Results, Latent Space Manipulations: Added explanation for the video in supplementary.
6. Experimental Results, edited and added FID report.
7. Appendix, Implementation Details: Explained the degradation process.
8. Appendix, added figures 9,10 for projection based editing and compositing
9. Appendix, added figure 15 describing the architecture.
10. Appendix, added figure 16, stress testing IGN $f^k(z) \quad k\rightarrow \infty$.
11. Appendix, added figure 17, illustration of gradients of idempotence loss.

---

### Public Comment · ~Michael_Maire1 · 2023-11-15
**Idempotent Generative Networks are a special case of EBGANs**

Contrary to this paper's claim of introducing a new approach to generative modeling, the proposed Idempotent Generative Network (IGN) is a special case of an Energy-Based Generative Adversarial Network (EBGAN) [Zhao, Mathieu, LeCun; ICLR 2017].  This connection appears to have been missed by both the authors and reviewers, as the paper does not cite, discuss, or compare to EBGAN, and none of the reviews mention it.  Though the related work section hints at the connection to GANs, the relationship is far stronger than "IGN incorporates elements of adversarial training."  IGN is exactly an EBGAN.

The functional form of IGN corresponds to the specific choice of setting the EBGAN generator and discriminator to utilize the same network (e.g., an autoencoder or U-Net architecture) and share parameters.  The three loss terms used in training an IGN then correspond exactly to the standard three components of GAN loss as implemented via MSE reconstruction loss in an EBGAN: (1) generator must fool discriminator (IGN's idempotent loss); (2) discriminator must identify real data (IGN's reconstruction loss); (3) discriminator must identify fake data (IGN's tight loss).  A derivation of this equivalence follows.

EBGAN is defined by a generator $G$, and a discriminator $D$, where $D$ is comprised of an encoder $Enc$ and a decoder $Dec$.  $D$ acts by examining reconstruction error of applying the autoencoder $Dec(Enc(y))$ to its input $y$.  The reconstruction error should be high for fake (generated) data and low for real data.  Training an EBGAN consists of alternatively training $D$ and $G$, according to objectives:

Training $D$:
$$
min_D [ ||Dec(Enc(x)) - x|| - ||Dec(Enc(G(z))) - G(z)|| ]
$$

Training $G$:
$$
min_G [ ||Dec(Enc(G(z))) - G(z)|| ]
$$

where $x$ denotes a real example and $z$ denotes a random input to the generator.

Now, choose $G$ to use the same network as the discriminator.  Specifically, choose to define:
$$
G(y) = Dec(Enc(y)) = f(y).
$$
Training this special case of EBGAN becomes:

Training $D$:
$$
min_{f_D} [ ||f_D(x) - x|| - ||f_D(f_G(z)) - f_G(z)|| ]
$$

Training $G$:
$$
min_{f_G} [ ||f_D(f_G(z)) - f_G(z)|| ]
$$

where $f_G(y) = f_D(y) = f(y)$ are all the same network, and subscripts $D$ and $G$ indicate which copies of $f$ are being trained in each step.  We can equivalently write this alternating training as one single procedure involving both steps (equivalent to alternating $D$ and $G$ training at the granularity of one example):

sample $x$

sample $z$

$$
\mathrm{L_rec}
   = ||f_D(x) - x||\ [\mathrm{training}\ f_D]
   = ||f(x) - x||
$$

$$
\mathrm{L_tight}
   = -||f_D(f_G(z)) - f_G(z)||\ [\mathrm{training\ only}\ f_D]
   = -||f(f(z).detach()) - f(z).detach()||
$$

$$
\mathrm{L_idem}
   = ||f_D(f_G(z)) - f_G(z)||\ [\mathrm{training\ only}\ f_G]
   = ||f_{copy}(f(z)) - f(z)||
$$

$$
\mathrm{L} = \mathrm{L_rec} + \mathrm{L_tight} + \mathrm{L_idem}
$$

These are precisely the losses in lines 15-17 of the IGN training routine.  IGN is exactly the special case of EBGAN wherein one chooses to implement the generator as being the same reconstruction network as the discriminator.

Thus, IGN is not a "new paradigm for generative modeling"; IGN is a specific form of EBGAN.  It is therefore not surprising that the paper remarks on mode collapse as a limitation of IGN.  IGN is not just similar to GANs; IGN is a GAN and hence inherits the limitations, such as mode collapse, often observed in GANs.

It is also not surprising that IGN, being an EBGAN, yields generation results whose quality are reminiscent of early work on GANs.  However, this cannot be interpreted as a promising preliminary result for a new paradigm.  Rather, IGN, itself a GAN, performing on par with earlier GANs is an entirely underwhelming result.  A convincing experimental case would need to demonstrate some advantage to making the particular choice to share network architecture and parameters between an EBGAN generator and discriminator.  Does such a choice improve generation quality over a baseline EBGAN that has an independently parameterized generator?  The paper makes no such analysis as it does not even recognize the connection to EBGAN.

---

> ### Author Response · Authors · 2023-11-15
>
> Thank you for your comment and insights.
> EBGAN is a very relevant work. We will add it to the paper and discuss it.
>
> Having the same model as both generator and discriminator is really the main point of IGN. The capabilities of IGN, mentioned in the paper and listed also in the responses here, all rise from exactly that and are not held by EBGAN. We briefly remind them and compare:
> 1. Sequential refinement of the model. EBGAN or any other GAN only do one step.
> 2. Projections of modified/corrupted images into the manifold. EBGAN does not demonstrate, and cannot trivially do that since it is not applied to images.
> 3. Editing of real images with no inversion, by simply applying the model to an input image. IGN gets images as inputs but EBGAN like other GANs cannot trivially do that.
> 4. Consistent latent space is the one property that we assume EBGAN also has, just like any other GAN. IGN has this consistency across sequential applications of the model.
>
> We should also point out that the strong relation to both GANs and EBMs is acknowledged in the related work section. (Though not mentioning specifically EBGAN).
> In particular, we describe the full analogy of IGN to GANs:
> "**One could view $\delta$ as a discriminator** trained using $L\_{rec}$ for real examples and $L\_{tight}$ for generated ones, while **$f$ serves as the generator** trained by $L\_{idem}$. Unique to IGN is a form of adversarial training we term “self-adversarial”, **Put simply, $f$ is both the generator and the discriminator**. This streamlined architecture affects the optimization process. Rather than alternating between two networks, a single model accumulates gradients from both discriminative and generative perspectives in each step."
> We also mention the relation to EBMs in which we acknowledge the similarity, and make the distinction that having an idempotent $f$ "reduces the need for separate optimization procedures to find the energy minimum".
>
> There are some other important implications to the difference, like the fact that there is no alternating optimization for IGN, but a single accumulation of the gradients from all losses. Another difference is the theoretical analysis. IGN has only one player, therefore cannot be formalized as a two-layer zero-sum game as GANs. Our theoretical derivation, while has similarities to that of GANs, is different.
>
> Over all, we appreciate this comment, and indeed EBGAN should have been discussed in the paper, and such a revision will be made soon.

---

> > ### Public Comment · ~Michael_Maire1 · 2023-11-23
> > **Re: Official Comment by Authors**
> >
> > The primary claim of the paper, that "[IGN] is a new paradigm for generative modeling" is false.  IGN is a GAN and the paper provides no evidence that it outperforms any other GAN variant in generation quality.
> >
> > Furthermore, the statements in the response above that, "there is no alternating optimization for IGN" and "IGN has only one player, therefore cannot be formalized as a two-player zero-sum game as GANs" are also false.  Please reread my derivation above, which shows the equivalence of both IGN's losses and IGN's optimization procedure to the adversarial EBGAN losses and the adversarial alternating EBGAN optimization procedure, respectively.
> >
> > Specifically, IGN's optimization is the limiting case of an EBGAN's alternating optimization between generator and discriminator when the alteration is done at the granularity of one example.  This alteration appears in IGN via usage of f_copy and f(z).detach().  That is, disconnecting gradients for these particular terms in the loss is precisely alternating between generator and discriminator training when taking a batch of two examples: one real ($x$) and one fake ($f(z)$).
> >
> > "Having the same model as both generator and discriminator is really the main point of IGN"
> >
> > A proper presentation of this idea would replace the entirety of Section 2 (Method) with the succinct explanation: "IGN is an EBGAN in which we have chosen to set $G(y) = Dec(Enc(y))$".  Instead, the method section of the paper consumes about two pages to, in a roundabout and somewhat obfuscated manner, derive training objectives that are, in fact, exactly the EBGAN losses, as well as an optimization procedure that is, in fact, exactly alternating generator and discriminator training one example at a time.
> >
> > To demonstrate that there is any point to IGN (EBGAN with generator = discriminator) would require showing that setting generator = discriminator in an EBGAN brings about capabilities that the vanilla EBGAN lacks.  The paper fails to establish such a claim, because it lacks experimental comparison to EBGAN.  Moreover, such experimental comparison is necessary because the standard EBGAN does train a network which maps image-like input to image-like output: the EBGAN $Dec(Enc())$ network, used by the discriminator, is precisely such a network (it has the form of IGN's desired mapping function $f()$).  What does this $Dec(Enc())$ network learn?  Certainly, one strategy consistent with being a good discriminator would be for the EBGAN $Dec(Enc())$ to learn a mapping onto the image manifold, in which case it might have many or all of the same capabilities of IGN's $f()$.
> >
> > A relevant quote from page 4 of the EBGAN paper addresses this possibility, "When trained with some regularization terms (see section 2.3.1), auto-encoders have the ability to learn an energy manifold without supervision or negative examples. This means that even when an EBGAN auto-encoding model is trained to reconstruct a real sample, the discriminator contributes to discovering the data manifold by itself."
> >
> > Thus, the claims in the above response about an EBGAN lacking the ability for "sequential refinement", "projection onto the manifold", and "editing of real images" have not been established; these are all open questions.  Answering them requires testing the capabilities of the $Dec(Enc())$ network trained in the standard EBGAN setting, where generator $G()$ is separately parameterized, against the capabilities of the $Dec(Enc())$ network trained in the IGN setting, where $G() = Dec(Enc())$.
> >
> > I strongly agree with Reviewer Cz9y's recent comment that, "this paper needs a major revision to incorporate EBGAN as a prior work and reframe its contributions."  The latest version of the IGN paper, which only adds an additional paragraph about EBGAN in the related work section, is an entirely insufficient response to comments that point out: (1) the method is not novel, but a special case of the widely known EBGAN paper from 2017, and (2) the proposed losses and optimization procedure are also not novel, but rather exactly equivalent to adversarial EBGAN training.  The paper's current claims and presentation do not meet the standards of scholarship that should be expected of ICLR publications.
> >
> > I would add that the paper also needs experimental comparison to an EBGAN baseline in order to answer the question of whether or not there is value to IGN.  Does constraining an EBGAN's generator to be the same network as its discriminator grant that discriminator any additional power?  One cannot simply declare this to be the case; this is an experimental question that requires experimental evidence.

---

> > > ### Author Response · Authors · 2023-11-23
> > >
> > > Thank you for sharing your insights.
> > > We appreciate your perspective on the relevance of EBGAN to IGN. We respectfully disagree with most of the points made and we will do our best to clarify them.
> > >
> > > The observation that IGN can be seen as a GAN with a single self-adversarial model was made in the paper, prior to this discussion.
> > >
> > > We believe using a single model is a non-trivial leap that indeed defines a new paradigm. To illustrate metaphorically, claiming that IGN is a special case of EBGAN is like claiming that $x\^2$ is a special case of linear function $ax$ with $a=x$ or that self-supervised learning is a special case of supervised learning. While such observations are useful in some specific cases, they do not capture the essence.
> > >
> > > We now respond to the specific comments, trying to clarify as much as possible:
> > >
> > > - "... no alternating optimization is false":
> > > IGN accumulates gradients from all losses and then the weights are updated using the sum of these gradients. Alternating would be the case if IGN updated the weights after back-propagating gradients from each loss separately.
> > >
> > > - "... cannot be formalized as a two-player zero-sum game as GANs are also false. Please reread my derivation above":
> > > The proposed derivation is showing equivalence to the loss terms if f=G=D. It does not prove convergence to target distribution in that case. Indeed, IGN is not a two-player game as there is only one model. This is beyond semantics. Theoretically proving convergence of IGN to the target distribution does not stem from GAN/EBGAN convergence. Plugging $G=D=f$ in a GAN convergence proof breaks it as it cannot be formulated as a min-max objective having the same model.
> > >
> > > - "replace the entirety of Section 2":
> > > IGN is derived from idempotence. It defines the motivation and describes it clearly, as acknowledged by some reviewers. The observation that it can be seen as a GAN with the same $f$ as both generator and discriminator comes later in the paper, because it does not hold a motivation. It is not clear why someone would look for some model which is self-adversarial, unless it rises from some other motivation. After discussion and clarification with the reviewers, we believe our derivation is both correct and motivated. Then, the relation to GANs does appear in the analysis. We believe it is better than the other way around (deriving IGN as a self-adversarial mechanism, then add analysis that it is idempotent).
> > >
> > > - "One strategy consistent with being a good discriminator would be for the EBGAN to learn a mapping onto the image manifold": There is no claim in the EBGAN paper or any evidence that a discriminator that was not trained for it, would learn to be idempotent or anything similar. This also does not seem likely. The discriminator only sees real or generated images. It never sees latents. We view this as quite speculative and perhaps an overly ambitious expectation to check that some existing models that never claimed to have some properties actually have them.
> > >
> > > - "EBGAN lacking the ability for sequential refinement, projection onto the manifold, and editing of real images have not been established": No experiment or claim in EBGAN suggests any of these properties. There is also no justification for them to exist. EBGAN uses a separate generator. The output of the discriminator is used to judge this result. We're unsure why one might assume that this discriminator can do sequential refinement, editing or projection. The EBGAN paper has no hint for any of these things. It's not immediately apparent to us how the quote brought from the EBGAN paper implies any of these.
> > >
> > > We summarize that while having a strong relation, IGN is different from EBGAN in every reasonable aspect.
> > > - **In practice:** There is only one model, no G and D competing. There is no alternating in training or separate optimizers.
> > > - **In theory:** Convergence of IGN is not given by GAN/EBGAN convergence as there is no min-max objective.
> > > - **In capabilities:** EBGAN doesn't demonstrate any of the listed advantages of IGN (Sequential refinement, Projection, Editing).
> > > - **In analysis:** Associating the idempotence property with generative modeling is not presented in EBGAN.
> > > - **In essence:** Even if we set aside all previous items, having the same $f$ with the same params to play both the generative and the discriminative roles, is in our view, a significant conceptual advancement.
> > >
> > >
> > > We appreciate this discussion and the insights by the commenter. We also thank reviewer Sz9m for participating and offering their perspectives in this discussion.

---

> > > > ### Public Comment · ~Michael_Maire1 · 2023-12-04
> > > > **Re: Official Comment by Authors**
> > > >
> > > > The above author response does not resolve the substance of my concerns. I maintain that: (1) The claim of a new generative paradigm is exaggerated; (2) IGN is a special case of an EBGAN and is a two-player game; (3) It is plausible that the EBGAN discriminator has capabilities the authors claim are exclusive to IGN. To determine whether IGN offers capabilities beyond EBGAN, a baseline experiment comparing to EBGAN is essential.
> > > >
> > > > **Motivation and Presentation**
> > > >
> > > > Claiming idempotency as motivation, the paper derives a system whose losses are identifiable with those of an EBGAN that uses the same function $f()$ as generator and discriminator. This is a different view of GANs, not a new generative paradigm. Moreover, if idempotency is a foundational principle from which one should derive such a GAN, why is there a loss ($L_\mathrm{tight}$) that explicitly pushes $f()$ away from idempotency while the other two losses ($L_\mathrm{rec}$, $L_\mathrm{idem}$) push it toward idempotency? Section 2.1's justification for $L_\mathrm{tight}$ does not follow from the principle of idempotency; a different reason is given for its development.
> > > >
> > > > If one instead starts from a GAN and considers using the same function for generator and discriminator, then the origin of all three loss terms immediately follows from the standard components of the GAN framework. Furthermore, idempotency of $f()$ is apparent as a consequence of training, provided the generator wins. This suggests that idempotency is more naturally viewed as a result of training a specific kind of GAN, rather than a foundational principle around which the framework is built.
> > > >
> > > > Compare to Consistency Models, which, although not presented as such, could be viewed as a generative framework motivated by idempotency. Consistency Models impose a boundary condition of idempotency on real data, and another loss term that encourages two inputs along the same noise trajectory to map to the same output. Unlike IGN, all losses involved in training Consistency Models are aligned with the concept of encouraging idempotency.
> > > >
> > > > **Two-Player Games**
> > > >
> > > > There is one model with two players adversarially manipulating that model's parameters. Each round of this two-player game is as follows:
> > > >
> > > > (1) Sample random data example $x$ and random noise $z$
> > > >
> > > > (2a) Player "G" computes loss $L_\mathrm{idem}$
> > > >
> > > > (2b) Player "D" computes loss $L_\mathrm{rec} + L_\mathrm{tight}$
> > > >
> > > > (3) Backprop from total loss and update parameters of $f()$
> > > >
> > > > Players "G" and "D" move (compute losses) in parallel. This minor difference from GANs is without consequence. Suppose we instead chose to backprop from loss $L_\mathrm{idem}$ and update parameters between steps (2a) and (2b), and then backprop from loss $L_\mathrm{rec} + L_\mathrm{tight}$ and update parameters after step (2b). Alternatively, suppose in a standard GAN we compute gradients for generator and discriminator in parallel. Whether the players make moves in series or parallel does not change the adversarial nature of the game. In either case, from one round to the next, each player sees a parameter update that is the combined consequence of their move and their opponent's move.
> > > >
> > > > **EBGAN Baseline**
> > > >
> > > > Any discriminator behaving as a score function could be retasked as a generator: iteratively update its input via gradient descent so as to maximize the realism score. Since the EBGAN discriminator is an autoencoder $Dec(Enc())$ and its score is the difference between input and output, instead of running gradient descent on the input with respect to the score, we can simply run the EBGAN discriminator in forward inference mode ($y \rightarrow Dec(Enc(y)$) to iteratively refine an image-like input $y$. This is the same functional form as IGN's $f()$.
> > > >
> > > > The GAN objectives place the following requirements on $Dec(Enc())$. First, $Dec(Enc(x)) = x$ for real samples $x$.  Second, $Dec(Enc(G(z)))$ must be different from $G(z)$. What is a consistent strategy that the EBGAN discriminator could learn for $Dec(Enc())$ in order to satisfy both? It could learn to map each input to the closest real sample! If it does, then it might also be capable of refinement, projection, and editing.
> > > >
> > > > IGN's $f()$ must satisfy the same two requirements, and a third: $f()$ must map a random noise input $z$ to a realistic example. Is this sufficient to make $f()$ more capable than $Dec(Enc())$? Unlike Diffusion Models or Consistency Models, $f()$ need not deal with arbitrary mixtures of data and noise; this extra modality does not expand the support of the input distribution in the same way. Other than the pure noise inputs, once generated examples become realistic, all inputs of $f()$ are close to the data distribution --- just like $Dec(Enc())$. Furthermore, we might expect a better discriminator to yield a better generator. $f()$ is both generator and discriminator, yet is no better as a generator than a standard GAN generator. Is $f()$ a better discriminator than EBGAN's $Dec(Enc())$?

---

> ### Comment · Reviewer_Sz9m · 2023-11-23
> **Comparisons to EBGAN**
>
> I would like to thank the Michael for their insights on the connections between EBGAN and IGN. However, I would also like to alert them, as they are a more senior researcher, that their tone has begun to become somewhat hostile, and I am beginning to become concerned that this might bring undo influence over the review process of this paper (particularly when received by more junior researchers).
>
> I intend to go over the connections between EBGAN and IGN (it's been a long time since I've read that paper) in detail, and I acknowledge the connections you have laid out. However, I may still consider the work to be publishable, as the insights are still interesting and novel.
>
> More specifically, I believe there are many ways to arrive at the same solution. And that's OK: these works should still be permitted to exist, despite they come around to a similar (or nearly the same) answer. If we are to completely discard such works, then we may unfortunately be stifling good and well-intentioned research, which *will reduce the quality of this conference*. Please keep this in mind: the purpose here should be to improve science, not gate-keep it.

---

### Meta-Review · Area_Chair_CEui · 2023-12-09

**Metareview:**

The paper proposes a way to train a new generative model class that encourages roughly three criteria: (1) For data points to be fixed points of a trained map f: that is f(x) = x for points x in the data distribution. (2) For the range of the source distribution z to be mapped to fixed points as well: that is f(f(z)) = f(z). (3) A "collapse-avoiding" term that avoids trivial solutions like f = identity. The authors provide convincing evidence that the trained models can perform various kinds of "denoising" by treating the learned f as a projector; they also provide a summary of a variety of training tricks to stabilize training.
I will mention there was a vigorous discussion with a public reviewer, as well as between the reviewers and myself. Ultimately the conclusion was that while in principle, the proposed method can be seen as an instantiation of EBGAN --- the mere fact that the method is related to / an instance of a prior method does not diminish a paper, so long as the paper is forthright about it. I am satisfied with the edits by the authors during the discussion period to point out the connection to EBGAN, as well as the authors discussion on Openreview laying out *specific* architectural choices, how to train them, how they are useful, and how they make certain tasks easier.

**Justification For Why Not Higher Score:**

I think the paper provides an interesting generative modeling paradigm, though arguably the experiments are not very extensive so the full extent of the benefit of the architecture is not completely clear. (The theory is not really a focal point of the paper and is more of a "sanity check" that in the limit of infinite capacity the objective makes sense.) I wouldn't be opposed to the paper having a spotlight, though, as the idea is clean and might be of wider interest.

**Justification For Why Not Lower Score:**

The paper has enough new ideas to be above the bar. The edits to the writing and the relationship to EBGAN in my view deal with the concerns of some of the reviewers and the public commenter.

---

### Decision · Program_Chairs · 2024-01-16

Accept (poster)